# `DFT2kp`: effective k · p models from *ab initio* data

João Victor V. Cassiano,[1,2] Augusto L. Araújo,[1,3] Paulo E. Faria Junior,[4] and Gerson J. Ferreira[1]

[1] *Instituto de Física, Universidade Federal de Uberlândia, Uberlândia, MG 38400-902, Brazil*

[2] *Instituto de Física, Universidade de São Paulo, São Paulo, SP, 05508-090, Brazil*

[3] *Ilum School of Science, CNPEM, C.P. 6192, 13083-970, Campinas, SP, Brazil*

[4] *Institute of Theoretical Physics, University of Regensburg, 93040 Regensburg, Germany*

The $\boldsymbol{k} \cdot \boldsymbol{p}$ method, combined with group theory, is an efficient approach to obtain the low energy effective Hamiltonians of crystalline materials. Although the Hamiltonian coefficients are written as matrix elements of the generalized momentum operator $\boldsymbol{\pi} = \boldsymbol{p} + \boldsymbol{p}_{\text{SOC}}$ (including spin-orbit coupling corrections), their numerical values must be determined from outside sources, such as experiments or *ab initio* methods. Here, we develop a code to explicitly calculate the Kane (linear in crystal momentum) and Luttinger (quadratic in crystal momentum) parameters of $\boldsymbol{k} \cdot \boldsymbol{p}$ effective Hamiltonians directly from *ab initio* wavefunctions provided by Quantum ESPRESSO. Additionally, the code analyzes the symmetry transformations of the wavefunctions to optimize the final Hamiltonian. This is an optional step in the code, where it numerically finds the unitary transformation $U$ that rotates the basis towards an optimal symmetry-adapted representation informed by the user. Throughout the paper, we present the methodology in detail and illustrate the capabilities of the code applying it to a selection of relevant materials. Particularly, we show a "hands-on" example of how to run the code for graphene (with and without spin-orbit coupling). The code is open source and available at https://gitlab.com/dft2kp/dft2kp.

## I. INTRODUCTION

The band structure of crystalline materials defines most of its electronic properties, and its accurate description is essential to the development of novel devices. For this reason, the *ab initio* density functional theory (DFT) [1, 2] provides one of the most successful tools for the development of electronics, spintronics, optoelectronics, etc. The DFT methods have been implemented in a series of codes (*e.g.*, Quantum ESPRESSO [3, 4], VASP [5], Wien2K [6], Gaussian [7], DFTB+ [8], Siesta [9, 10], ...), which differ by the choice of basis functions (*e.g.*, localized orbitals or plane-waves), pseudo-potential approximations, and other functionalities. Nevertheless, all DFT implementations provide methods to obtain the equilibrium (relaxed) crystalline structure, phonon dispersion, and electronic band structures. Complementary, few bands effective models are essential to further study transport, optical, and magnetic properties of crystalline materials. These can be developed either via the tight-binding (TB) [11–13] or $\boldsymbol{k} \cdot \boldsymbol{p}$ method [14, 15], which complement each other.

On the one hand, the TB method has an "atomistic" nature, since it is built upon localized basis sets (*e.g.*, maximally-localized Wannier functions [16], or atomic orbitals), which makes this method optimal for numerical modeling of transport, optical and other properties of complex *nanomaterials* [17–20].

On the other hand, the $\boldsymbol{k} \cdot \boldsymbol{p}$ method uses basis sets of extended waves, which are exact solutions of the Hamiltonian at a quasi-momentum of interest, typically at a high symmetry point of the Brillouin zone. While this characteristic may limit the $\boldsymbol{k} \cdot \boldsymbol{p}$ description to a narrow region of the energy-momentum space, the $\boldsymbol{k} \cdot \boldsymbol{p}$ Hamiltonians are easier to handle analytically and, especially, are very suitable to study mesoscopic systems using the envelope function approximation [21–26]. For example, the k.p framework has been successfully applied to study nanostructures (quantum wells, wires, and dots) [27–29], topological insulators [30–32], spin-lasers [33, 34], polytypism [35–37], as well as a large variety of two-dimensional van der Waals materials [38–41]. Moreover, recent developments in the field of transition metal dichalcogenides (TMDCs) have combined DFT and $\boldsymbol{k} \cdot \boldsymbol{p}$ methodologies to explore the valley Zeeman physics in TMDC monolayers and their van der Waals heterostructures [42–45].

Both the TB and $\boldsymbol{k} \cdot \boldsymbol{p}$ Hamiltonians are defined in terms of arbitrary coefficients. In the TB case, these are local site energies and hopping amplitudes described by Slater-Koster matrix elements [11]. For the $\boldsymbol{k} \cdot \boldsymbol{p}$ Hamiltonians, these are the Kane [46, 47] and Luttinger [48] parameters, which are matrix elements of the momentum and spin-orbit coupling operators. In both methods (TB or $\boldsymbol{k} \cdot \boldsymbol{p}$), the values of these arbitrary coefficients must be determined from outside sources, which strongly depend on the size and analytical properties of the particular model Hamiltonian. For instance, early studies within the $\boldsymbol{k} \cdot \boldsymbol{p}$ framework have shown that for parabolic single band descriptions, or weakly coupled models, it is possible to write the quadratic coefficients in terms of effective masses, which can be experimentally determined by cyclotron resonance experiments [46, 49–52]. Moreover, energy splittings, such as band gaps, can be directly determined from optical experiments [53–57]. For III-V semiconductors with zinc-blend structure and nitride-based wurtzite compounds, a useful database for $\boldsymbol{k} \cdot \boldsymbol{p}$ parameters inspired by experimentally available datasets can be found in Ref. [58]. Conversely, for $\boldsymbol{k} \cdot \boldsymbol{p}$ Hamiltonians that do not allow analytical solutions, but still have a low number of bands ($\sim 10$), it is possible to perform numerical fitting techniques to DFT calculations [39, 41, 59–65]. For larger $\boldsymbol{k} \cdot \boldsymbol{p}$ Hamiltonians ($> 30$ bands), fitting proce-

arXiv:2306.08554v2 [cond-mat.mes-hall] 7 Feb 2024

dures may also be applied [66, 67] or directly extracted from first principles calculations, since the only matrix elements involved are linear in momentum [68–70]. Interestingly, these large band $\boldsymbol{k} \cdot \boldsymbol{p}$ models can even be used to supplement and speed up first principles calculations, as demonstrated in Refs. [68–70]. In TB models, fitting procedures can also be applied to obtain the unknown parameters [71–75]. Conversely, fully automated procedures, integrated within *ab initio* codes, such as the `wannier90` code [76, 77], use localized Wannier functions computed from the DFT wave functions to calculate TB parameters. Moreover, explicit calculations of the Slater-Koster matrix elements are implemented in the `paoflow` [78] and DFTB+ [8] codes.

While it is possible to extract $\boldsymbol{k} \cdot \boldsymbol{p}$ models from a Taylor expansion on top of a TB model (*e.g.,* via the code `tbmodels` [79]), there are no versatile implementations to calculate the $\boldsymbol{k} \cdot \boldsymbol{p}$ Kane (linear in k) and Luttinger (quadratic in $\boldsymbol{k}$) parameters directly from the DFT wavefunctions [80]. To calculate the $\boldsymbol{k} \cdot \boldsymbol{p}$ matrix elements from the DFT wavefunctions, one needs to account for how the wavefunctions are represented in the DFT code [69, 81]. For instance, Quantum ESPRESSO and VASP implement pseudopotential approximations within the Projector Augmented Wave (PAW) method [82–85]. Fortunately, Quantum ESPRESSO already provides a routine to calculate matrix elements of the velocity operator (which is sufficient to obtain $\boldsymbol{k} \cdot \boldsymbol{p}$ models, as we see in Section B). Indeed, recently, Jocić and collaborators [86] have successfully calculated $\boldsymbol{k} \cdot \boldsymbol{p}$ models directly from QE's wavefunctions (see disclaimer at our Conclusions).

In this paper, we present an *open-source code* that automatically calculates the numerical values for the $\boldsymbol{k} \cdot \boldsymbol{p}$ Kane and Luttinger parameters using the wavefunctions provided by Quantum ESPRESSO (QE). For this purpose, first, we develop a patch to instruct QE to calculate and store the matrix elements of the generalized momentum $\boldsymbol{\pi} = \boldsymbol{p} + \boldsymbol{p}_{\mathrm{SOC}}$, which includes the spin-orbit corrections. Together with the eigenenergies $E_n^0$ at $\boldsymbol{k}_0$, the matrix elements of $\boldsymbol{\pi}$ for a selected set of $N$ bands define the effective $\boldsymbol{k} \cdot \boldsymbol{p}$ Hamiltonian $H_{N \times N}(\boldsymbol{k})$ for $\boldsymbol{k}$ near $\boldsymbol{k}_0$. Our `python` package reads these matrix elements and QE's wavefunctions $|n\rangle$ to automatically build $H_{N \times N}(\boldsymbol{k})$ using Löwdin's partitioning [87] for the folding down of all QE bands into the selected $N$ bands subspace. Additionally, the user has the option to improve the appearance (or *form*) of the effective Hamiltonian via a symmetry optimization process aided by the `qsymm` package [88], which builds the symbolic Hamiltonian via group theory and the method of invariants. To illustrate the capabilities of our code, we show here a step-by-step "hands-on" tutorial on how to run the code for graphene, and later we present results for selected materials [zincblende, wurtzite, rock-salt, transition metal dichalcogenides (TMDC), and others]. In all cases, the modeled band structure matches remarkably well the DFT data at low energies near the expansion point $\boldsymbol{k}_0$. Our code is open source and available at the `gitlab` repository [89].

This paper is organized as follows. In Section II we present our methodology starting with a brief review of the $\boldsymbol{k} \cdot \boldsymbol{p}$ method, Löwdin partitioning, the method of invariants, the symmetry optimization process, and the calculation of matrix elements using the DFT data. Next, in Section III, we show the code in detail using graphene as a practical example. Later, in Section IV, we illustrate the results of the code for zincblend (GaAs, CdTe, HgTe), wurtzite (GaP, GaN, InP), rock-salt (SnTe, PbSe), a TMDC (MoS$_2$), and other materials (Bi$_2$Se$_3$, GaBiCl$_2$). We finish the paper with an overview of the results in Section V, and the conclusions.

## II. METHODS

Our goal is to obtain the numerical values for the coefficients of $\boldsymbol{k} \cdot \boldsymbol{p}$ effective Hamiltonians [14, 15]. Namely, these are the Kane [46, 47] and Luttinger [48] parameters. To present our approach to this calculation, let us start by briefly describing its fundamental steps. First, we review the $\boldsymbol{k} \cdot \boldsymbol{p}$ method to show that these coefficients depend only upon matrix elements of the type $\boldsymbol{P}_{m,n} = \langle m| \boldsymbol{\pi} |n\rangle$, where $\boldsymbol{\pi} = \boldsymbol{p} + \boldsymbol{p}_{\mathrm{SOC}}$ is the generalized momentum operator with the spin-orbit corrections, and $\{|n\rangle\}$ is the set of numerical wavefunctions obtained from the *ab initio* DFT simulations (e.g., via Quantum ESPRESSO [3, 4]). However, the numerical DFT basis given by $\{|n\rangle\}$ does not match, *a priori*, the optimal symmetry-adapted basis set that yields the desired form for the effective $\boldsymbol{k} \cdot \boldsymbol{p}$ Hamiltonian. Therefore, to properly identify the Kane and Luttinger parameters, we perform a *symmetry optimization*, which rotates the arbitrary numerical basis into the optimal symmetry-adapted form. This symmetry optimization is performed via group theory [90, 91] by enforcing that the numerical DFT basis transforms under the same representation of an *optimal symmetry-adapted basis*, which is informed by the user.

In summary, the algorithm steps are:

1. Read the QE/DFT data: energies $E_n^0$ and eigenstates $|n\rangle$ at the selected $\boldsymbol{k}_0$ point.

2. Calculate or read the matrix elements of $\boldsymbol{P}_{m,n} = \langle m| \boldsymbol{\pi} |n\rangle$ for all bands $(m, n)$.

3. Select the bands of interest (set $A$). The code will identify the irreducible representations of the bands using the `IrRep` python package [92], and present it as a report to the user. Additionally, the code calculates the model folded down into the selected set $A$ via Löwdin partitioning.

4. Build the optimal effective model from symmetry constraints using the `Qsymm` python package [88] under an optimal symmetry-adapted basis informed by the user. This optimal basis must be in a set of

representations equivalent to the ones identified in Step 3.

5. Calculate the representation matrices for the symmetry operators in the original QE basis $|n\rangle$. The code verifies if the representations of the numerical QE basis are equivalent to the representations of the optimal symmetry-adapted basis from step 4.

6. Calculates the transformation matrix $U$ that rotates the original QE basis into the optimal symmetry-adapted basis set in step 4. Applies the transformation $U$ and calculates the optimal symmetry-adapted numerical effective Hamiltonian.

7. Convert values from Rydberg atomic units into meV and nm units, and present a report with values for the $\boldsymbol{k} \cdot \boldsymbol{p}$ parameters.

In the next sections, we describe the relevant details of the steps above, but not following the algorithmic order above. More specifically, in Section II A, we briefly review the $\boldsymbol{k} \cdot \boldsymbol{p}$ formalism to show that $\boldsymbol{P}_{m,n} = \langle m| \boldsymbol{\pi} |n\rangle$ plays a central role in our approach. Incidentally, we introduce the folding down via Löwdin partitioning [87]. Next, we define what is the optimal symmetry-adapted form of the Hamiltonian via the method of invariants [15, 93] in Section II B. In Section II C, we present the symmetry optimization approach to calculate the transformation matrix $U$ that yields our final $H^{\text{optimal}} = U \cdot H^{\text{DFT}} \cdot U^{\dagger}$. At last, in Section II D we discuss how $\boldsymbol{P}_{m,n} = \langle m| \boldsymbol{\pi} |n\rangle$ is calculated.

Throughout the paper we use atomic Rydberg units (a.u.), thus the reduced Planck constant, bare electron mass and charge are $\hbar = 2m_0 = e^2/2 = 1$, the permittivity of vacuum is $4\pi\varepsilon_0 = 1$, the speed of light is $c = 2/\alpha \approx 274$, and $\alpha \approx 1/137$ is the fine structure constant.

## A. The k · p model

In this section, we briefly review the $\boldsymbol{k} \cdot \boldsymbol{p}$ method [14, 15, 46–48] and the folding down via Löwdin partitioning [15, 87, 93] to establish our notation.

We are interested in the effective Hamiltonian near a high-symmetry point $\boldsymbol{k}_0$ of the Brillouin zone. Therefore, we write the quasi-momentum as $\boldsymbol{\kappa} = \boldsymbol{k}_0 + \boldsymbol{k}$, such that $\boldsymbol{k}$ is the deviation from $\boldsymbol{k}_0$. The Bloch theorem allow us to decompose the wavefunction as $\psi_{\boldsymbol{\kappa}}(\boldsymbol{r}) = e^{i\boldsymbol{k}\cdot\boldsymbol{r}}\phi_{\boldsymbol{k}_0,\boldsymbol{k}}(\boldsymbol{r})$, with $\phi_{\boldsymbol{k}_0,\boldsymbol{k}}(\boldsymbol{r}) = e^{i\boldsymbol{k}_0\cdot\boldsymbol{r}}u_{\boldsymbol{k}_0+\boldsymbol{k}}(\boldsymbol{r})$, where $u_{\boldsymbol{k}_0+\boldsymbol{k}}(\boldsymbol{r}) \equiv u_{\boldsymbol{\kappa}}(\boldsymbol{r})$ is the periodic part of the Bloch function, while $\phi_{\boldsymbol{k}_0,\boldsymbol{k}}(\boldsymbol{r})$ carries the phase given by $\boldsymbol{k}_0$ and obeys the Schrödinger equation $[H^0 + H'(\boldsymbol{k})]\phi_{\boldsymbol{k}_0,\boldsymbol{k}}(\boldsymbol{r}) = [E - k^2]\phi_{\boldsymbol{k}_0,\boldsymbol{k}}(\boldsymbol{r})$, with

$$H^0 = p^2 + V(\boldsymbol{r}) + 2\boldsymbol{k}_0 \cdot \boldsymbol{\pi} + H_{\text{SR}}, \qquad (1)$$

$$H'(\boldsymbol{k}) = 2\boldsymbol{k} \cdot \boldsymbol{\pi}, \qquad (2)$$

$$\boldsymbol{\pi} = \boldsymbol{p} + \frac{\alpha^2}{8}\boldsymbol{\sigma} \times \nabla V(\boldsymbol{r}), \qquad (3)$$

where $H^0$ is the Hamiltonian at $\boldsymbol{k} = 0$, $V(\boldsymbol{r})$ is the periodic potential, $H'(\boldsymbol{k})$ carries the k-dependent contributions that will be considered as a perturbation hereafter, $\boldsymbol{\pi}$ is the generalized momentum that includes the spin-orbit contributions (SOC), and $\boldsymbol{\sigma} = (\sigma_x, \sigma_y, \sigma_z)$ are the Pauli matrices for the electron spin. For simplicity, we consider only leading order corrections of the fine structure terms. Namely, at $\boldsymbol{k} = 0$, the $H_{\text{SR}}$ carries the scalar relativistic terms, composed by the Darwin, $H_{\text{D}} = \frac{\alpha^2}{8}\nabla^2 V(\boldsymbol{r})$, and the mass-velocity corrections, $H_{\text{MV}} = -\alpha^2 p^4/4$. In the *ab initio* DFT data, these are implied in the numerical eigenvalues $E_n^0$ of $H^0$. For finite $\boldsymbol{k} \neq 0$, we keep only the SOC contribution in $\boldsymbol{\pi}$, and neglect the higher order mass-velocity corrections (see Appendix A).

The DFT data, as shown in the next section, provide us with a set $\{|n\rangle\}$ of eigenstates of $H^0$, i.e. $H^0 |n\rangle = E_n^0 |n\rangle$. From this *crude DFT basis*, we define an all bands model $H_{\text{all}}^{\text{DFT}}(\boldsymbol{k})$, with matrix elements

$$\langle m| H_{\text{all}}^{\text{DFT}} |n\rangle = E_n^0 \delta_{m,n} + 2\boldsymbol{k} \cdot \boldsymbol{P}_{m,n}, \qquad (4)$$

where $\boldsymbol{P}_{m,n} = \langle m| \boldsymbol{\pi} |n\rangle$. We refer to this as the *crude model* because it is calculated from the original numerical DFT wavefunctions, which do not have an optimal symmetry-adapted form (more detail in Section II C). Nevertheless, it already shows that $E_n^0$ and $\boldsymbol{P}_{m,n}$ are central quantities, and both can be extracted from DFT simulations, as shown in Section II D.

Next, we want to fold down $H_{\text{all}}^{\text{DFT}}$ into a subspace of $N$ bands near the Fermi energy to obtain our reduced, but still crude, effective model $H_{N\times N}^{\text{DFT}}$. This is done via Löwdin partitioning [15, 87, 93]. First, the user must inform the set of $N$ bands of interest, which we refer to as set $A$. Complementary, the remaining remote bands compose the set $B$. Considering the diagonal basis $H^0 |n\rangle = E_n^0 |n\rangle$, and the perturbation $H'(\boldsymbol{k})$, the Löwdin partitioning leads to the effective Hamiltonian $H_{N\times N}^{\text{DFT}}$ defined by the expansion

$$[H_{N\times N}^{\text{DFT}}]_{m,n}(\boldsymbol{k}) = \left(E_n^0 + k^2\right)\delta_{m,n} + H'_{m,n}(\boldsymbol{k})$$

$$+ \frac{1}{2}\sum_{r\in B} H'_{m,r}(\boldsymbol{k})H'_{r,n}(\boldsymbol{k})\left(\frac{1}{E_m^0 - E_r^0} + \frac{1}{E_n^0 - E_r^0}\right) + \cdots \qquad (5)$$

with $H'_{m,n}(\boldsymbol{k}) = \langle m| H'(\boldsymbol{k}) |n\rangle = 2\boldsymbol{k} \cdot \boldsymbol{P}_{m,n}$. Here, the indices $m, n \in A$ run over the bands we want to model (set $A$), while $r \in B$ run over the remote bands. The expansion above is shown up to second order in $H'$, but higher order terms can be found in Ref. [15]. Alternatively, the recent python package `pymablock` [94] implements an efficient numerical method to compute the Löwdin partitioning to arbitrary order.

## B. The optimal symmetry-adapted form of H

The selection rules from group theory allow us to identify which matrix elements of an effective Hamiltonian are finite [90]. More interestingly, the method of invariants [15, 93] can be used to directly obtain the most general form of $H_{N \times N}^{\text{optimal}}(\boldsymbol{k})$ allowed by symmetry. To define this form, consider a Taylor series expansion

$$H_{N \times N}^{\text{optimal}}(\boldsymbol{k}) = \sum_{i,j,l} h_{i,j,l} \; k_x^i \; k_y^j \; k_z^l, \tag{6}$$

where $h_{i,j,l}$ are constant matrices that multiply the powers of $\boldsymbol{k} = (k_x, k_y, k_z)$ as indicated by its indices $i, j, l = \{0, 1, 2, \dots\}$. To find the symmetry allowed $h_{i,j,l}$, we recall that the space group $\mathcal{G}$ of the crystal is defined by symmetry operations that keep the crystalline structure invariant. Particularly, at a high symmetry point $\boldsymbol{\kappa} = \boldsymbol{k}_0$, one must consider the little group $\mathcal{G}_{\boldsymbol{k}_0} \in \mathcal{G}$ of symmetry operations that maintain $\boldsymbol{k}_0$ invariant (the star of $\boldsymbol{k}_0$). Hence, $H_{N \times N}^{\text{optimal}}(\boldsymbol{k})$ must commute with the symmetry operations of $\mathcal{G}_{\boldsymbol{k}_0}$. Namely,

$$H_{N \times N}^{\text{optimal}}(D^k(S)\boldsymbol{k}) = D^\psi(S) H_{N \times N}^{\text{optimal}}(\boldsymbol{k}) D^\psi(S^{-1}), \tag{7}$$

where $D^\psi(S)$ are the representation matrices for each symmetry operator $S \in \mathcal{G}_{\boldsymbol{k}_0}$ in the subspace defined by the wavefunctions of set $A$, and $D^k(S)$ are the representation matrices acting on the vector $\boldsymbol{k} = (k_x, k_y, k_z)$. The set of equations defined by this relation for all $S \in \mathcal{G}_{\boldsymbol{k}_0}$ leads to a linear system of equations that constrain the symmetry allowed form of $H_{N \times N}^{\text{optimal}}(\boldsymbol{k})$, i.e., it defines which of constant matrices $h_{i,j,l}$ are allowed up to a multiplicative factor. Ultimately, these multiplicative factors are the Kane and Luttinger parameters that we want to calculate numerically.

The python package Qsymm [88] implements an efficient algorithm to find the form of $H_{N \times N}^{\text{optimal}}(\boldsymbol{k})$ solving the equation above and returns the symmetry allowed $h_{i,j,l}$. Qsymm refers to these as the *Hamiltonian family*. To perform the calculation, the user must inform the representation matrices $D^\psi(S)$ for the generators of $\mathcal{G}_{\boldsymbol{k}_0}$. Notice that the choice of representation is arbitrary, and different choices lead to effective Hamiltonians with different forms. This ambiguity is the reason the next step, symmetry optimization, is necessary.

## C. Symmetry optimization

In the previous section, the matrix representations for generators $S \in \mathcal{G}_{\boldsymbol{k}_0}$ are implicitly written in an *optimal symmetry-adapted basis*, which we will now label with an $\mathcal{O}$ index, as in $\{|n_{\mathcal{O}}\rangle\}$, to distinguish from the *crude DFT* numerical basis, which we now label with an $\mathcal{C}$ index, as in $\{|n_{\mathcal{C}}\rangle\}$. The matrix representations of $S$ written in these two bases are equivalent up to a unitary transformation $U$, i.e. $D^{\mathcal{O}}(S) = U \cdot D^{\mathcal{C}}(S) \cdot U^\dagger$. Indeed, this same matrix $U$ transforms the *crude DFT* numerical Hamiltonian into the desired *optimal symmetry-adapted form*, i.e. $H_{N \times N}^{\text{optimal}} = U \cdot H_{N \times N}^{\text{DFT}} \cdot U^\dagger$. Therefore, our goal here is to find this transformation matrix $U$.

For each symmetry operator $S_i \in \mathcal{G}_{\boldsymbol{k}_0}$, let us define $\mathcal{C}^i \equiv D^{\mathcal{C}}(S_i)$ and $\mathcal{O}^i \equiv D^{\mathcal{O}}(S_i)$ as the representation matrices under the original numerical DFT basis ($\mathcal{C}$), and under the desired optimal symmetry-adapted representation ($\mathcal{O}$), respectively. For irreducible representations, this $U$ is unique (modulo a phase factor) and an efficient method to obtain it was recently developed [95] and used in Ref. [86] to transform the effective model into the desired form. The procedure described in Ref. [95] is exact but relies on a critical step where one has to find for which indices $(a, b)$ the weight matrix $r_{a,b}$ is finite. For transformations between irreps, any of the finite $r_{a,b}$ lead to equivalent unitary transformations. However, for transformations between reducible representations, one needs to identify, within the set of finite $r_{a,b}$, the ones that yield nonequivalent transformation matrices that combine to form the final transformation matrices $U$. This can be a complicated numerical task. Here, instead, we propose an alternative method that applies more easily to reducible representations and allows us to obtain the transformation matrix $U$ with a systematic approach. Next, we describe the method, and later in Sec. III C we illustrate its capabilities using the spinful graphene example.

The set of unitary transformations $\mathcal{O}^i = U \cdot \mathcal{C}^i \cdot U^\dagger$ for each $S_i \in \mathcal{G}_{\boldsymbol{k}_0}$ compose a system of equations for $U$. These can be written in terms of its matrix elements in a linearized form that reads as

$$\sum_j U_{m,j} \mathcal{C}_{j,n}^i - \mathcal{O}_{m,j}^i U_{j,n} = 0. \tag{8}$$

Defining a vector $\boldsymbol{V} = \{U_{1,1}, U_{1,2}, \cdots, U_{2,1}, \cdots, U_{N,N}\}^T$, where $N$ is the order of the representations (number of bands in set $A$), allow us to cast the equation above as $\boldsymbol{Q}_i \cdot \boldsymbol{V} = 0$, with $\boldsymbol{Q}_i = 1_N \otimes (\mathcal{C}^i)^T - \mathcal{O}^i \otimes 1_N$ of size $N^2 \times N^2$, and $1_N$ as the $N \times N$ identity matrix. Since the same similarity transformation $U$ must apply for all $S_i$, we stack each $\boldsymbol{Q}_i$ into a rectangular matrix $\boldsymbol{Q} = [\boldsymbol{Q}_1, \boldsymbol{Q}_2, \cdots, \boldsymbol{Q}_q]^T$ of size $(qN^2) \times N^2$. The full set of equations now read as $\boldsymbol{Q} \cdot \boldsymbol{V} = 0$, such that the solution $\boldsymbol{V} = \sum_{j=1}^{N_Q} c_j \boldsymbol{v}_j$ is a linear combination of the nullspace $\{\boldsymbol{v}_j\}$ of $\boldsymbol{Q}$, with coefficients $c_j$ and nullity $N_Q$. The matrix $U$ can be recovered from the elements of $\boldsymbol{V}$, which follow from its definition above. If $u_j$ is the matrix reconstructed form of $\boldsymbol{v}_j$, we can write $U = \sum_{j=1}^{N_Q} c_j u_j$.

Additionally, it is interesting to consider anti-unitary symmetries. These can be either the time-reversal symmetry (TRS) itself, or combinations of TRS and space group operations (magnetic symmetries) [90, 91]. For instance, in spinful graphene neither TRS nor spatial inversion are symmetries of the K point, but their composition is an important symmetry that enforces a constraint on the allowed SOC terms (see Sec. III C). Following a notation similar to the one above, let us refer

to these magnetic symmetries as $\bar{\mathcal{C}}^i = D^{\mathcal{C}}(\bar{S}_i)\mathcal{K} \equiv \tilde{\mathcal{C}}^i\mathcal{K}$ and $\bar{\mathcal{O}}^i = D^{\mathcal{O}}(\bar{S}_i)\mathcal{K} \equiv \tilde{\mathcal{O}}^i\mathcal{K}$, where $\mathcal{K}$ is the complex conjugation, and $(\tilde{\mathcal{C}}^i, \tilde{\mathcal{O}}^i)$ are the unitary parts of $(\bar{\mathcal{C}}^i, \bar{\mathcal{O}}^i)$. Now the basis transformation for these symmetries read as $\tilde{\mathcal{O}}^i = U^* \cdot \tilde{\mathcal{C}}^i \cdot U^\dagger$, where we choose to apply $\mathcal{K}$ to the left (this choice is for compatibility with the python package IrRep [92]). To add this equation to the $\boldsymbol{Q}$ matrix above, we consider $U$ and $U^*$ as independent variables. Then, as above, it follows the linearized form

$$\sum_j U^*_{m,j}\tilde{\mathcal{C}}^i_{j,n} - \tilde{\mathcal{O}}^i_{m,j}U_{j,n} = 0. \tag{9}$$

In all cases, the expression for the transformation matrix is $U = \sum_{j=1}^{N_Q} c_j u_j$, where the coefficients $c_j$ are so far undefined. To find these coefficients $c_j$, we numerically minimize the residues $R(\{c_j\}) = \sum_i ||\mathcal{O}_i - U \cdot \mathcal{C}^i \cdot U^\dagger||^2$, and $\tilde{R}(\{c_j\}) = \sum_i ||\tilde{\mathcal{O}}_i - U^* \cdot \tilde{\mathcal{C}}^i \cdot U^\dagger||^2$. The global minima of these residues, $R(\{c_j\}) = \tilde{R}(\{c_j\}) \equiv 0$, yields $a$ solution $U(\{c_j\})$, such that small perturbations to the coefficients $c_j \to c_j + \delta c_j$ lead to quadratic deviations from the minima, $e.g.$, $R \propto |\delta c_j|^2$. This procedure opens a question of whether the solution $U(\{c_j\})$ at the global minima is unique.

Since $U$ represents a transformation between two basis sets ($e.g.$, $|n_{\mathcal{O}}\rangle = U|n_{\mathcal{C}}\rangle$), it expected to be unique. However, the problem here is formulated such that we explicitly have the eigenstates $|n_{\mathcal{C}}\rangle$ that compose the crude DFT basis set $\mathcal{C}$, while for the optimal symmetry-adapted basis set $\mathcal{O}$ we know only how we expect the eigenstates $|n_{\mathcal{O}}\rangle$ to transform under the symmetry operations of the group. Therefore, instead of solving for $U$ directly from the linear basis transformation $|n_{\mathcal{O}}\rangle = U|n_{\mathcal{C}}\rangle$, we rely on the quadratic equations for the transformation between the symmetry operators ($e.g.$, $D^{\mathcal{O}}(S) = U \cdot D^{\mathcal{C}}(S) \cdot U^\dagger$), or their linearized forms in Eq. (8) and Eq. (9). First, consider that $\mathcal{O}$ and $\mathcal{C}$ refer to distinct, but equivalent irreps. As emphasized in [95], it follows from Schur's lemma that the transformation $U$ is unique modulo a phase. Indeed, for the unitary constraints, $\mathcal{O}^i = U \cdot \mathcal{C}^i \cdot U^\dagger$, the solution $U$ is invariant under $U \to e^{i\theta}U$ for any real $\theta$, while for the anti-unitary constraint, $\tilde{\mathcal{O}}^i = U^* \cdot \tilde{\mathcal{C}}^i \cdot U^\dagger$, $U$ is invariant only for $\theta = 0$ or $\pi$. Next, without loss of generality, let us consider that $\mathcal{O}$ and $\mathcal{C}$ refer to reducible representations already cast in block-diagonal forms. In this case, the solution $U = U_1 \oplus U_2 \oplus \cdots$ also takes a block-diagonal form, where each block $U_j$ corresponds to a transformation within a single irrep subspace. It follows that each $U_j$ is unique modulo the phases above. The overall global phase of $U$ does not affect the calculation of our matrix elements. However, the arbitrary relative phases between the blocks $U_j$ might lead to ill-defined phases of matrix elements between eigenstates of different irreps if the anti-unitary symmetries are not informed. In contrast, if anti-unitary symmetries are used, the undefined phase factor in the matrix elements is just a sign.

## D. Matrix elements via DFT

As shown above, our approach to obtain a $\boldsymbol{k} \cdot \boldsymbol{p}$ model directly from the DFT data relies on two quantities: (i) the band energies $E_n^0$ at the $\boldsymbol{k} \cdot \boldsymbol{p}$ expansion point $\boldsymbol{k}_0$; and (ii) the matrix elements $\boldsymbol{P}_{m,n} = \langle m|\boldsymbol{\pi}|n\rangle$ also calculated at $\boldsymbol{k}_0$ for all bands $\{|n\rangle\}$. The band energies $E_n^0$ are a straightforward output of any DFT code. Therefore, here we discuss only the calculation of $\boldsymbol{P}_{m,n} = \langle m|\boldsymbol{\pi}|n\rangle$.

We focus on the Quantum ESPRESSO (QE) [3, 4] implementation of $ab$ $initio$ DFT [1, 2]. There, the Hamiltonian is split into the core and intercore regions via the Projector Augmented Wave (PAW) method [82–84], which is backward compatible with ultrasoft (USPPs) [83, 96] and norm-conserving pseudo-potentials (NCPP) [97–99]. In these approaches, the atomic core region is replaced by pseudopotentials, which are constructed from single-atom DFT simulations with the Dirac equation in the scalar relativistic or full relativistic approaches. Thus, for molecules or crystals, QE solves a pseudo-Schrödinger equation, with the atomic potentials replaced by the pseudopotentials. Here we shall not go through the details of the PAW and pseudopotential methods. For the interested reader, we suggest Refs. [82–84]. Instead, for now, it is sufficient to conceptually understand that QE provides numerical solutions for the Schrödinger equation with the fine structure corrections, which can be expressed by the Hamiltonian

$$H \approx p^2 + V(\boldsymbol{r}) + H_{\mathrm{SR}} + \frac{\alpha^2}{4}(\boldsymbol{\sigma} \times \nabla V) \cdot \boldsymbol{p}, \tag{10}$$

where $H_{\mathrm{SR}} = H_{\mathrm{D}} + H_{\mathrm{MV}}$ contain the Darwin and mass-velocity contributions, as presented above, and the last term is the spin-orbit coupling.

### 1. Matrix elements of the velocity

Fortunately, the QE code already provides tools to calculate the matrix elements of the velocity operator $\frac{1}{2}\boldsymbol{v} = \frac{i}{2}[H, \boldsymbol{r}]$, which reads as

$$\frac{\boldsymbol{v}}{2} = \frac{1}{2}\frac{\partial H}{\partial \boldsymbol{p}} = \boldsymbol{\pi} + \frac{1}{2}\frac{\partial H_{\mathrm{MV}}}{\partial \boldsymbol{p}} \approx \boldsymbol{\pi}, \tag{11}$$

where we neglect the mass velocity corrections (see Appendix A). Thus, we find that $\boldsymbol{P}_{m,n} = \langle m|\boldsymbol{\pi}|n\rangle \approx \langle m|\frac{1}{2}\boldsymbol{v}|n\rangle$. The calculation of $\boldsymbol{P}_{m,n}$ is already partially included in the post-processing tool `bands.x` (file `PP/src/bands.f90`), within the `write_p_avg` subroutine (file `PP/src/write_p_avg.f90`). This calculation includes the necessary PAW, USPPs, or NCPPs corrections, which are critical for materials where the wavefunction strongly oscillates near the atomic cores [100]. However, the `write_p_avg` subroutine only calculates $|\boldsymbol{P}_{m,n}|^2$ for $m$ in the valence bands (below the Fermi level) and $n$ in the conduction bands (above the Fermi level). To overcome this limitation, we have built a patch that modifies

bands.f90 and write_p_avg.f90 to calculate $\boldsymbol{P}_{m,n}$ for all bands. This leads to a modified bands.x with options to follow with its original behavior or to calculate $\boldsymbol{P}_{m,n}$ according to our needs. This is controlled by a new flag lpall = False/True added to the input file of bands.x in addition to the lp = True. Its default value (lpall = False) runs bands.x with its original code, while the option lpall = True instructs bands.x to store all $\boldsymbol{P}_{m,n}$ into the file indicated by the input parameter filp.

In general, it is preferable to patch QE to use the full $\boldsymbol{P}_{m,n}$, since the calculation is faster and more precise. Nevertheless, if the user prefers not to apply our patch to modify QE, our code can calculate an approximate $\boldsymbol{P}_{m,n}$ using only the plane-wave components outputted by the QE code. In this case, we consider that the pseudo-wavefunction is a reasonable approximation for the all-electron wavefunction, thus neglecting PAW corrections, which are necessary to account for SOC. Therefore, under this approximation, $\boldsymbol{P}_{m,n} \approx \langle m | \boldsymbol{p} | n \rangle$. The relevance of these PAW/SOC corrections to $\boldsymbol{P}_{m,n}$ are presented in the example shown in Sec. IV B 1. Within this approximation, the wavefunction $\psi_{n,\boldsymbol{k}}(\boldsymbol{r})$ for the band $n$ at quasi-momentum $\boldsymbol{k}$, and $\boldsymbol{P}_{m,n}$ read as

$$\psi_{n,\boldsymbol{k}}(\boldsymbol{r}) \approx \frac{1}{\sqrt{\Omega}} \sum_{\boldsymbol{G}} c_n(\boldsymbol{G}) e^{i(\boldsymbol{k}+\boldsymbol{G})\cdot\boldsymbol{r}}, \qquad (12)$$

$$P_{m,n} \approx \sum_{\boldsymbol{G}} (\boldsymbol{k}+\boldsymbol{G}) c_m^\dagger(\boldsymbol{G}) c_n(\boldsymbol{G}), \qquad (13)$$

where $c_n(\boldsymbol{G})$ are the plane-wave expansion coefficients (spinors in the spinful case), $\Omega$ is the normalization volume, and $\boldsymbol{G}$ are the lattice vectors in reciprocal space. To implement this calculation, and the one shown next, we use the IrRep python package [92], since it already has efficient routines to read and manipulate the QE data.

### 2. Matrix elements of the symmetry operators

To calculate the matrix elements of the symmetry operators, it is sufficient to consider $\psi_{n,\boldsymbol{k}}(\boldsymbol{r})$ from Eq. (12). In this case, it is safe to neglect PAW corrections, since they must transform identically to the plane-wave parts under the symmetry operations of the crystal space group. For a generic symmetry operation $S \in \mathcal{G}_{\boldsymbol{k}_0}$, its matrix elements read as

$$D_{m,n}^\psi(S) = \sum_{\boldsymbol{G},\boldsymbol{G}'} c_m^\dagger(\boldsymbol{G}') c_n(\boldsymbol{G})$$
$$\int e^{-i(\boldsymbol{k}+\boldsymbol{G}')\cdot\boldsymbol{r}} e^{-iS^{-1}(\boldsymbol{k}+\boldsymbol{G}')\cdot\boldsymbol{r}} \frac{d^3r}{\Omega}. \quad (14)$$

Using the plane-wave orthogonality, one gets

$$D_{m,n}^\psi(S) = \sum_{\boldsymbol{G}} c_m^\dagger\big(-\boldsymbol{k}+S^{-1}\cdot(\boldsymbol{k}+\boldsymbol{G})\big) c_n(\boldsymbol{G}), \quad (15)$$

where $S^{-1}$ is the inverse of $S$, and $S^{-1} \cdot (\boldsymbol{k}+\boldsymbol{G})$ is its action on the $(\boldsymbol{k}+\boldsymbol{G})$ vector. For instance, if $S = I$ is

the spatial inversion symmetry, $S^{-1} \cdot (\boldsymbol{k}+\boldsymbol{G}) = -\boldsymbol{k}-\boldsymbol{G}$, and $D_{m,n}^\psi(S) = \sum_{\boldsymbol{G}} c_m^\dagger(-2\boldsymbol{k}-\boldsymbol{G}) c_n(\boldsymbol{G})$.

## III. HANDS-ON EXAMPLE: GRAPHENE

In this section, we present a detailed example and results for spinless graphene, and a shorter discussion on spinful graphene in Sec. III C to illustrate the case of transformations between reducible representations. Graphene [101, 102] is nowadays one of the most studied materials due to the discovery of its Dirac-like effective low energy model, which reads as $H = \hbar v_F \boldsymbol{\sigma} \cdot \boldsymbol{k}$. Here, the $\boldsymbol{\sigma}$ Pauli matrices act on the orbital pseudo-spin subspace, $\boldsymbol{k} = (k_x, k_y)$ is the quasi-momentum, and $v_F$ is the Fermi velocity, which is the unknown coefficient that we want to calculate in this example. For this purpose, we follow a pedagogical route in this first example. First, we present the symmetry characteristics of the graphene lattice and its wavefunctions at the K point. Then, we show the results for the representation matrices and Hamiltonian in the crude and optimal symmetry-adapted basis to illustrate how the symmetry optimization of Section II C is used to build the optimal symmetry-adapted Hamiltonians and identify the numerical values for its coefficients. Later, in Section III B we show a step-by-step tutorial on how to run the code. This example was chosen for its simplicity, which allows for a clear discussion of each step. Later, in Section IV we present a summary of examples for other materials of current interest.

Before discussing the details, we summarize the results for the band structure of graphene in Fig. 1, which compares the DFT data with our two main models. The black lines are calculated from the all bands model from Eq. (4), which uses the matrix elements $\boldsymbol{P}_{m,n}$ in the original crude DFT basis without further processing. In contrast, the red lines are the band structure calculated with the folded-down Hamiltonian for a set $A$ composed by the two bands near the Fermi energy that defines the Dirac cone, and considers the symmetry optimization process to properly identify the $\boldsymbol{k} \cdot \boldsymbol{p}$ parameters. This optimal symmetry-adapted Hamiltonian is shown in Eq. (21) below, and the numerical value for its parameters is shown at *Step 7* in Section III B.

### A. Overview of the theory and symmetry optimization

The crystal structure of graphene is a hexagonal monolayer of carbon atoms, as shown in Figs. 1(a) and 1(b), which is invariant under the P6/mmm space group (#191). However, since its Dirac cone is composed of $p_z$ orbitals only, it is sufficient to consider the $C_{6V}$ factor group to describe the lattice. Particularly, at the K point [see Fig. 1(c)], the star of K corresponds to the little group $C_{3V}$, which is generated by a 3-fold rotation $C_3(z)$ and a mirror $M_y$. The Dirac bands of graphene are

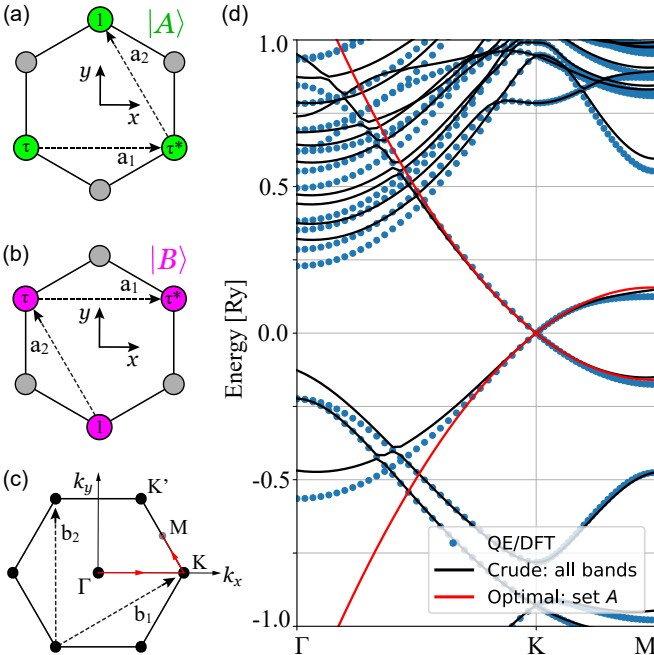

Figure 1. Graphene lattices emphasizing the Dirac cone eigenstates at the K point, where (a) $|A\rangle = |(X + iY)Z\rangle$ and (b) $|B\rangle = |(X - iY)Z\rangle$. Both eigenstates are composed by $p_z$ orbitals centered at the colored sites (A and B lattices) with the Bloch phase factors indicated within the circles, where $\tau = \exp(i2\pi/3)$. (c) The first Brillouin zone, marking the path $\Gamma - K - M$ used to plot the bands in (d). (d) Band structure for graphene calculated via QE/DFT (blue circles), all bands model [Eq. (4)] (black lines), and optimal symmetry-adapted model [Eq. (21)] for the two bands forming the Dirac cone (red). Here, the QE/DFT simulation was performed with 300 bands.

characterized by the irrep $E$ of $C_{3V}$ (or irrep $K_6$ from P6/mmm [103]), which is composed by basis functions $(xz, yz)$.

To build the optimal symmetry-adapted effective model via the method of invariants, we need to specify a basis and calculate the matrix representation of the symmetry operations mentioned above. Since the wavefunctions of the Dirac cone transform as the irrep $E$ of $C_{3V}$, a naive choice would be $A_{\text{unconv}} = \{|XZ\rangle, |YZ\rangle\}$, which corresponds to a set $A$ in Section II A. This choice of basis refers to a possible $\mathcal{C}$ representation in Section II C, and it yields

$$D^{\text{unconv}}(C_3(z)) = \begin{pmatrix} \cos\theta & -\sin\theta \\ \sin\theta & \cos\theta \end{pmatrix}, \tag{16}$$

$$D^{\text{unconv}}(M_y) = \begin{pmatrix} 1 & 0 \\ 0 & -1 \end{pmatrix}, \tag{17}$$

$$H_{\text{unconv}} \approx \begin{pmatrix} c_0 - c_1 k_x & c_1 k_y \\ c_1 k_y & c_0 + c_1 k_x \end{pmatrix}, \tag{18}$$

where $\theta = 2\pi/3$. Here $H_{\text{unconv}}$ is obtained via Qsymm up to linear order in $\mathbf{k}$, for brevity. While the eigenenergies of $H_{\text{unconv}}$ represent correctly the Dirac cone as

$E_\pm = c_0 + |c_1|\sqrt{k_x^2 + k_y^2}$, the Hamiltonian $H_{\text{unconv}}$ takes an undesirable unconventional form.

A more convenient choice is $A_{\text{conv}} = \{|(X + iY)Z\rangle, |(X - iY)Z\rangle\}$, which is illustrated in Figs. 1(a) and 1(b). This choice of basis leads to

$$D^{\text{conv}}(C_3(z)) = \begin{pmatrix} e^{i\theta} & 0 \\ 0 & e^{-i\theta} \end{pmatrix}, \tag{19}$$

$$D^{\text{conv}}(M_y) = \begin{pmatrix} 0 & 1 \\ 1 & 0 \end{pmatrix}, \tag{20}$$

$$H_{\text{conv}} \approx \begin{pmatrix} c_0 & c_1 k_- \\ c_1 k_+ & c_0 \end{pmatrix} + \begin{pmatrix} c_2 k^2 & c_3 k_+^2 \\ c_3 k_-^2 & c_2 k^2 \end{pmatrix}, \tag{21}$$

where $k_\pm = k_x \pm i k_y$. Now, up to linear order in k, we see that $H_{\text{conv}} \approx c_0 + c_1 \boldsymbol{\sigma} \cdot \mathbf{k}$, where $\boldsymbol{\sigma}$ act on the subspace set by $A_{\text{conv}}$, and we identify $c_1 = \hbar v_F$. Additionally, the k-quadratic terms that lead to trigonal warping corrections. Notice that both choices, $A_{\text{unconv}}$ and $A_{\text{conv}}$, are equivalent representations, but the conventional one leads to the familiar *form* of the graphene Hamiltonian. These two basis sets are related by an unitary transformation $U$, such that $A_{\text{conv}} = U \cdot A_{\text{unconv}}$ and $H_{\text{conv}} = U H_{\text{unconv}} U^\dagger$, with

$$U_{\text{unconv} \to \text{conv}} = \frac{1}{\sqrt{2}} \begin{pmatrix} 1 & i \\ 1 & -i \end{pmatrix}. \tag{22}$$

Next, let us analyze the set $A_{\text{QE}}$ of numerical wavefunctions from QE. Do they correspond to $A_{\text{QE}} = A_{\text{conv}}$ or $A_{\text{QE}} = A_{\text{unconv}}$? The answer is neither. Since it is a raw numerical calculation, typically diagonalized via the Davidson algorithm [104], a degenerate or nearly degenerate set of eigenstates might be in any linear combination of its representative basis. Therefore, the symmetry optimization step is essential to find the matrix transformation $U$ that yields $A_{\text{conv}} = U \cdot A_{\text{QE}}$. To visualize this, let us check the matrix representations of the symmetry operators above, and the effective Hamiltonian calculated from the crude QE data. For the symmetry operators, we find

$$D^{\text{QE}}(C_3(z)) \approx \begin{pmatrix} -0.5 & -0.35 + 0.79i \\ 0.35 + 0.79i & -0.5 \end{pmatrix}, \tag{23}$$

$$D^{\text{QE}}(M_y) \approx \begin{pmatrix} +0.5 & 0.35 - 0.79i \\ 0.35 + 0.79i & -0.5 \end{pmatrix}, \tag{24}$$

While this cumbersome numerical representation does not resemble neither $A_{\text{conv}}$ nor $A_{\text{unconv}}$, our symmetry optimization process correctly finds a transformation matrix $U$ that returns $A_{\text{conv}} = U \cdot A_{\text{QE}}$, where

$$U \approx \begin{pmatrix} 0.7i & -0.28 + 0.65i \\ -0.6 + 0.37i & 0.7 - 0.1i \end{pmatrix}. \tag{25}$$

Finally, for the Hamiltonian, up to linear order in k and in the original QE basis, we find

$$H_{\mathrm{QE}} \approx \begin{pmatrix} -0.37 & -0.25 + 0.57i \\ -0.25 - 0.57i & 0.37 \end{pmatrix} k_x$$
$$+ \begin{pmatrix} 0.62 & 0.15 - 0.34i \\ 0.15 + 0.34i & 0.62 \end{pmatrix} k_y, \quad (26)$$

which takes a cumbersome *form* in this raw numerical basis. However, applying the transformation $U$, the symmetry adapted model becomes

$$H_{N \times N}^{\mathrm{optimal}} = U H_{N \times N}^{\mathrm{DFT}} U^{\dagger} \approx 0.72\, \boldsymbol{\sigma} \cdot \boldsymbol{k}. \quad (27)$$

Here we identify $\hbar v_F = 0.72$ in Rydberg units, yielding $v_F = 0.83 \times 10^6$ m/s. The resulting band structure calculated from $H^{\mathrm{optimal}}$, including the k-quadratic terms, is shown as red lines in Fig. 1(d) and it matches well the QE/DFT data near K.

## B.  Running the code

The example presented here is available in the `Examples/graphene-nosoc.ipynb` notebook in the code repository, and shown in Algorithm 1.   Here we show only the minimal procedure to read the DFT data, build an effective model from the symmetry constraints, and calculate the numerical values for the model parameters.  Complementary, the full code in `Examples/graphene-nosoc.ipynb` shows how to plot the data presented in our figures.

For now, we assume that the DFT simulation was successful. The suggested steps to run QE and prepare the data for our code is to run the `calculation='scf'` and `calculation='bands'` with `pw.x`.   Then, run `bands.x` to extract the bands from QE's output and store it in `gnuplot` format to plot the figures. Here, for graphene, we assume that the `bands` calculation was run for a path $\Gamma - \mathrm{K} - \mathrm{M}$ with 30 points between each section, such that K is the 31st point in the list.

Next, we describe each step shown in Algorithm 1.

*Step 1.*   After running QE, the first step is to read the DFT data from the QE's output folder. The command `dft2kp.irrep(...)` uses the python package `IrRep` [92] to read the data for the selected k point to be used in the $\boldsymbol{k} \cdot \boldsymbol{p}$ expansion, as indicated by the parameters `kpt` and `kname`. The data is read from the folder indicated by the parameter `dftdir`, while `outdir` and `prefix` refer to values used in the input file of QE's `pw.x` calculation. Additionally, the command `dft2kp.irrep(...)`  also accepts extra parameters from the package `IrRep` (see code documentation).

*Step 2.*   In step 2, the code will either read or calculate the matrix elements $\boldsymbol{P}_{m,n}$ to build the effective models. If the user runs QE modified by our patch, the QE tool `bands.x` will generate a file `kp.dat` that already contains the values for $\boldsymbol{P}_{m,n}$. In this case, the user must inform

the name of this file via the parameter `qekp`. Otherwise, if `qekp` is omitted, our code calculates an approximate value for $\boldsymbol{P}_{m,n} \approx \langle m| \boldsymbol{p} |n\rangle$ from the pseudo-wavefunction of QE, as in Eq. (13), which neglects all SOC corrections.

*Step 3.*   Next, the user must choose which set of bands will be considered to build the model. This is the set $A$ in Section II A. In this example, we select bands 3 and 4, which correspond to the Dirac cone of graphene. The code analyzes the list of bands and identifies their irreducible representations (irreps) using the `IrRep` package [92]. Here, the set $A$ must contain only complete sets of irreps, otherwise the Löwdin perturbation theory would fail with divergences [see Eq. (5)], since the remote bands of set $B$ would have at least one band degenerated with a band from set $A$. If this condition fails, the code stops with an error message. Otherwise, if set $A$ is valid, the code outputs a report indicating the space group of the crystal (e.g., P6/mmm), the selected set of bands (e.g., [3,4]), their irrep (e.g., $K_6$ [103]), and degeneracy (2). The report reads as

```
Space group  191: P6/mmm
Verifying set A: [3 4]
Band indices: [3, 4] Irreps: (K6) Degeneracy: 2
```

Additionally, in this step, the code also calculates the crude effective model for the bands in set $A$ via Löwdin partitioning [87]. It stores the folded Hamiltonian in a Python dictionary (`kp.Hdict`) representing the matrices $h_{i,j,l}$ in the crude DFT basis that define $H^{\mathrm{DFT}}(\boldsymbol{k}) = \sum_{i,j,l} h_{i,j,k} k_x^i k_y^j k_z^l$.   For instance, `kp.Hdict['xx']` refers to the matrix $h_{2,0,0}$ that defines the term $h_{2,0,0} k_x^2$.

*Step 4.*   In step 4 we build the optimal symmetry-adapted model using `Qsymm` [88], which solves Eq. (7) for the method of invariants. In Algorithm 1, we build the representations for the symmetry operations $C_3(z)$, $M_y$, $M_z$, and $\mathcal{TI}$. Above we have discussed only the first two for simplicity. Here we also include the mirror $M_z$, and the anti-unitary symmetry $\mathcal{TI}$, which is composed of the product of time-reversal and spatial inversion symmetries. The mirror $M_z$ has a trivial representation $D^{\psi}(M_z) = -1$, since the orbitals that compose the Dirac bands in graphene are all of Z-like (odd in z). The $\mathcal{TI}$ representation follows from $A_{\mathrm{conv}}$ presented above by recalling that spinles time-reversal is simply the complex conjugation and the spatial inversion takes $(X, Y, Z) \rightarrow (-X, -Y, -Z)$. In this particular example, the $\mathcal{TI}$ symmetry does not play an important role, but it is essential for a spinful graphene example, as it constrains the SOC terms at finite $\boldsymbol{k}$ (see Sec. III C). The command `dft2kp.qsymm(...)`  calls `Qsymm` to build the effective model from the list of symmetries, indicated by `symm`, up to order $k^2$, as indicated by `total_power`. We recommend always using `dim=3` [three dimensions for $\boldsymbol{k} = (k_x, k_y, k_z)$] because QE always work with the 3D space groups. Additionally, the command

`dft2kp.qsymm(...)` accepts other parameters that are given to the `Qsymm` package (see code documentation). By default, this command outputs the optimal symmetry-adapted Hamiltonian, which matches the one in Eq. (21).

*Step 5.* Next, we start the symmetry optimization process. The first call `kp.get_symm_matrices()` calculates, via Eq. (15), the matrix representation for all symmetry operators identified in the QE data by the `IrRep` package. However, neither QE nor `IrRep` account for the anti-unitary symmetries. Therefore, we call here the optional routine `kp.add_antiunitary_symm(...)`, which manually adds the anti-unitary symmetry to the list of QE symmetries and matches it with the corresponding symmetry of `Qsymm` informed on its first parameter. In this example, we add the $\mathcal{TI}$ symmetry built with `Qsymm` above. This operator needs to be complemented with a possible non-symmorphic translation vector, which is zero in this case, as shown by the second parameter of `kp.add_antiunitary_symm(...)`. Both calls, `kp.get_symm_matrices()` and `kp.add_antiunitary_symm(...)`, calculate the matrix representations in the crude QE basis.

*Step 6.* To calculate the transformation matrix $U$, we compare the ideal matrix representations informed via `Qsymm` (object `qs`) and the crude QE matrix representations (object `kp`). The call `dft2kp.basis_transform(...)` performs this comparison and returns an error if the symmetries in both objects do not match. More importantly, it calculates the transformation matrix $U$ solving Eq. (8) and Eq. (9). The matrix $U$ is stored in the object `optimal.U`. If the calculation of $U$ is successful, the code applies $U$ to rotate the $h_{i,j,l}$ terms in `kp.Hdict` from the crude DFT basis into the optimal symmetry-adapted basis. This allows for direct identification of the coefficients $c_n$ from Eq. (21), which are stored in `optimal.coeffs`. Additionally, the code builds the numerical optimal symmetry-adapted model and provides a callable object `optimal.Heff(kx, ky, kz)` that returns the numerical Hamiltonian $H_{N \times N}^{\mathrm{optimal}}$ for a given value of $\boldsymbol{k} = (k_x, k_y, k_z)$.

*Step 7.* At last, the code prints a report with the numerical values for the coefficients $c_n$, which are summarized in Table I. As mentioned above, here we identify $\hbar v_F = 0.72$ a.u., yielding $v_F = 0.83 \times 10^6$ m/s after converting the units.

**Algorithm 1** Minimal example for spinless graphene.

```python
import numpy as np
import pydft2kp as dft2kp

# import s0, sx, sy, sz: Pauli matrices
from pydft2kp.constants import s0, sx, sy, sz

# step 1: read DFT data
kp = dft2kp.irrep(dftdir='graphene-nosoc',
                  outdir='outdir',
                  prefix='graphene',
                  kpt=31,
                  kname='K')

# step 2: read or calculate matrix elements of p
kp.get_p_matrices(qekp='kp.dat')

# step 3: define the set alpha
#         applies fold down via Löwdin
setA = [3, 4]
kp.define_set_A(setA)

# step 4: builds optimal model with qsymm
phi = 2*np.pi/3
U = np.diag([np.exp(1j*phi), np.exp(-1j*phi)])
C3 = dft2kp.rotation(1/3, [0,0,1], U=U)
My = dft2kp.mirror([0,1,0], U=sx)
Mz = dft2kp.mirror([0,0,1], U=-s0)
TI = dft2kp.PointGroupElement(R=-np.eye(3),
                              conjugate=True,
                              U=sx)
symms = [C3, My, Mz, TI]
qs = dft2kp.qsymm(symms, total_power=2, dim=3);

# step 5: calculate the representation matrices
kp.get_symm_matrices()
# (optional): adds anti-unitary symmetry
kp.add_antiunitary_symm(TI, np.array([0,0,0]))

# step 6: calculates and applies
#         the transformation U
optimal = dft2kp.basis_transform(qs, kp)

# step 7: print results
optimal.print_report(sigdigits=3)
```

Table I. Graphene parameters for the Hamiltonian of Eq. (21).

| Coefficient | Values in a.u. | Values in (eV, nm) |
|:-----------:|:--------------:|:------------------:|
| $c_0$ | $\sim 0$ | $\sim 0$ eV |
| $c_1$ | 0.72 | 0.52 eV nm |
| $c_2$ | $\sim 0$ | $\sim 0$ eV nm$^2$ |
| $c_3$ | 0.82 | 0.031 eV nm$^2$ |

### C. Spinful graphene

To complement the example above, we consider now the spinful graphene (full code available at `Examples/graphene.ipynb` [89]). In this case, due to the small spin-orbit coupling of graphene, the numerical DFT basis functions from QE mix two nearly degenerate irreps into an unintended reducible representation. Nevertheless, our symmetry optimization procedure can properly block diagonalize the symmetry operators according to the intended representation.

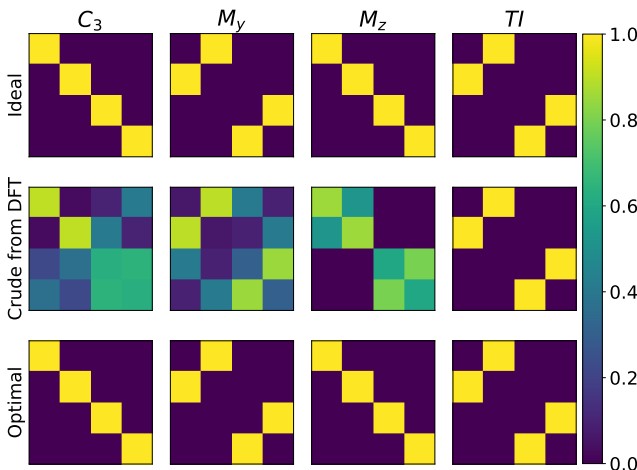

Figure 2. The absolute value of the representation matrices of the symmetry operations for the spinful graphene example, as labeled on top of each column. The top line of matrices are defined under the ideal basis informed by the user, i.e. $\{|(X+iY)Z,\uparrow\rangle, |(X-iY)Z,\downarrow\rangle, |(X-iY)Z,\uparrow\rangle, |(X+iY)Z,\downarrow\rangle\}$, as discussed in the text. The central line shows the calculated representation matrices under the crude DFT basis from QE, which does not split into the ideal block-diagonal form due to the small SOC gap between the bands. Applying our transformation $U$ to the crude representation from the central line, we obtain the optimal symmetry-adapted basis that lead to the proper block-diagonal form of the representation matrices shown in the bottom line.

To see this, let us first establish the ideal basis in proper ordering that leads to the block-diagonal form of the symmetry operators $C_3(z)$, $M_y$, $M_z$, and $\mathcal{TI}$ (considering the group generators only). Thus, considering the spin, the basis functions now read as $\{|(X+iY)Z,\uparrow\rangle, |(X-iY)Z,\downarrow\rangle, |(X-iY)Z,\uparrow\rangle, |(X+iY)Z,\downarrow\rangle\}$. Under the P6/mmm double space group [103, 105], this set of basis functions transform as the sum of two bidimensional irreps [106], namely $\bar{K}_7 \oplus \bar{K}_9$. Under this basis, the symmetry operators listed above take a block-diagonal form, which are illustrated in the top row of Fig. 2. Algebraically, these read

$$D^{\text{ideal}}(C_3) = \begin{pmatrix} -\tau^* & 0 & 0 & 0 \\ 0 & -\tau & 0 & 0 \\ 0 & 0 & -1 & 0 \\ 0 & 0 & 0 & 1 \end{pmatrix}, \qquad (28)$$

$$D^{\text{ideal}}(M_y) = \begin{pmatrix} 0 & -1 & 0 & 0 \\ 1 & 0 & 0 & 0 \\ 0 & 0 & 0 & -1 \\ 0 & 0 & 1 & 0 \end{pmatrix}, \qquad (29)$$

$$D^{\text{ideal}}(M_z) = \begin{pmatrix} i & 0 & 0 & 0 \\ 0 & -i & 0 & 0 \\ 0 & 0 & i & 0 \\ 0 & 0 & 0 & -i \end{pmatrix}, \qquad (30)$$

$$D^{\text{ideal}}(\mathcal{TI}) = \begin{pmatrix} 0 & 1 & 0 & 0 \\ -1 & 0 & 0 & 0 \\ 0 & 0 & 0 & 1 \\ 0 & 0 & -1 & 0 \end{pmatrix} K. \qquad (31)$$

In contrast to the block diagonal form of the $D^{\text{ideal}}(\cdots)$ matrices above, the representation matrix for the $C_3(z)$ calculated with the crude DFT basis from QE takes the form

$$D^{\text{QE}}(C_3) \approx$$
$$\begin{pmatrix} -0.9-0.1i & -0.0-0.0i & +0.1+0.1i & -0.3-0.3i \\ +0.0-0.0i & -0.9+0.1i & -0.3+0.3i & -0.1+0.1i \\ -0.2-0.1i & +0.0-0.4i & +0.4+0.5i & +0.3+0.6i \\ +0.0+0.4i & +0.2-0.1i & -0.3+0.6i & +0.4-0.5i \end{pmatrix}.$$
$$(32)$$

Similarly, the crude DFT representation for $M_y$, $M_z$ and $\mathcal{TI}$ also show non-block-diagonal forms in the central line of Fig. 2.

The algorithm described in Sec. II C builds a system of equations to find the transformation matrix $U$ that yields $D^{\text{ideal}}(S) = UD^{\text{QE}}(S)U^\dagger$ for all symmetry $S$ of the group (i.e., $S = \{C_3(z), M_y, M_z, \mathcal{TI}\}$ in this example). The Python code to implement this procedure is nearly identical to Algorithm 1, requiring only (i) the expansion of `setA`, in *Step 3*, to account for the 4 bands that compose the spinful Dirac cone (i.e., `setA = [6, 7, 8, 9]` in this Example); and (ii) the replacement of the symmetry matrices from *Step 4* for the ones listed above. From these, in *Step 6* we find the transformation matrix

$$U \approx$$
$$\begin{pmatrix} +0.1-0.0i & -0.1+0.2i & -0.6-0.6i & -0.4-0.2i \\ +0.1-0.2i & -0.0-0.1i & +0.4+0.2i & -0.9-0.0i \\ +0.2+0.2i & -0.9+0.1i & -0.1+0.2i & +0.0+0.1i \\ -0.6-0.7i & -0.3+0.0i & -0.1+0.1i & +0.1-0.2i \end{pmatrix},$$
$$(33)$$

which precisely yields the transformation $UD^{\text{QE}}(S)U^\dagger = D^{\text{optimal}}(S) \equiv D^{\text{ideal}}(S)$, as illustrated in the bottom row of Fig. 2.

The model resulting from the considerations above read as

$$H_{\text{sfg}} = \begin{pmatrix} c_0 & 0 & -c_2 k_- & 0 \\ 0 & c_0 & 0 & -c_2 k_+ \\ -c_2 k_+ & 0 & c_1 & 0 \\ 0 & -c_2 k_- & 0 & c_1 \end{pmatrix}$$
$$+ \begin{pmatrix} c_4 k^2 & 0 & -c_5 k_+^2 & 0 \\ 0 & c_4 k^2 & 0 & -c_5 k_-^2 \\ -c_5 k_-^2 & 0 & c_6 k^2 & 0 \\ 0 & -c_5 k_+^2 & 0 & c_6 k^2 \end{pmatrix}, \quad (34)$$

where $k^2 = k_x^2 + k_y^2$, $k_\pm = k_x \pm i k_y$, and we omit $k_z$-dependent for 2D materials. Notice that if we do not consider the composed magnetic anti-unitary symmetry $\mathcal{TI}$, the $c_2$ and $c_5$ terms above split into real and imaginary parts. Particularly for $c_2$, the real part refers to matrix elements of $\boldsymbol{p}$, while the imaginary part would carry contributions from $\boldsymbol{p}_{\text{soc}}$. Nevertheless, considering $\mathcal{TI}$, these coefficients are expected to be real and the $\boldsymbol{p}_{\text{soc}}$ contributions to the imaginary part vanish by symmetry.

The numerical values found for the parameters of $H_{\text{sfg}}$ in Eq. (34) are shown in Table II. The Fermi velocity matches the one from spinless graphene above, and we find that the intrinsic spin-orbit coupling is $\lambda_I = c_1 - c_0 \approx 1$ $\mu$eV, which is much smaller than its established value of $\lambda_I \approx 24$ $\mu$eV obtained via all-electron full-potential DFT implementations [107, 108]. This discrepancy is due to limitations of the pseudo-potentials used here with QE [109], which do not include d orbitals. Nevertheless, this example serves to show that, whenever two irreps are nearly degenerate, the DFT wavefunctions might always be mixed into reducible representations and the symmetry optimization procedure implemented here efficiently rotates the DFT basis back into ideal form that yields block-diagonal reducible representations.

Table II. Spinful graphene parameters for the Hamiltonian of Eq. (34).

| Coefficient | Values in a.u. | Values in (eV, nm) |
|---|---|---|
| $c_0$ | $-1.39 \times 10^{-5}$ | $-0.000189$ eV |
| $c_1$ | $-1.40 \times 10^{-5}$ | $-0.000190$ eV |
| $c_2$ | $0.72$ | $0.518$ eV nm |
| $c_4$ | $0.049$ | $0.0018$ eV nm$^2$ |
| $c_5$ | $-0.82$ | $-0.031$ eV nm$^2$ |
| $c_6$ | $0.049$ | $0.0018$ eV nm$^2$ |

## IV. EXAMPLES

In this section, we briefly show the results for a series of selected materials without presenting a step-by-step tutorial as above. More details for each case below can be seen in the code repository. Here we consider examples of zincblende crystals (GaAs, HgTe, CdTe), wurtzite crystals (GaN, GaP, InP), rock-salt crystals (SnTe, PbSe), a transition metal dichalcogenide monolayer (MoS$_2$), 3D and 2D topological insulators (Bi$_2$Se$_3$, GaBiCl$_2$). Additional examples can be found in the code repository. In all cases, the resulting models agree well with the DFT bands near the $\boldsymbol{k} \cdot \boldsymbol{p}$ expansion point and low energies, as expected. The DFT parameters used in the simulations are presented in Appendix B.

### A. Zincblende crystals

We consider well-known zincblende crystals: GaAs, CdTe and HgTe. These crystals are characterized by lattices that transform as the space group F$\bar{4}$3m, but their low energy bandstructure concentrates near the $\Gamma$ point, which can be described by the point group $T_d$ after factorizing the invariant subgroup of Bloch translations. The basis functions and effective Kane model for these materials are well described in the literature [14, 15, 91]. Here, let us simply summarize this characterization to establish a notation.

In all cases considered in this section, the first conduction band and the top valence bands transform either as $S$ or $P = (X, Y, Z)$ orbitals, and in terms of the crystallographic coordinates we define $x \parallel [100]$, $y \parallel [010]$, and $z \parallel [001]$. In the single group $T_d$, neglecting spin, the S-like orbitals transform accordingly to the trivial $A_1$ irrep of $T_d$, while the P-like orbitals transform as the $T_2$ irrep. Including spin, the double group representation for the S-like orbitals become $A_1 \otimes D_{1/2} = \bar{\Gamma}_6$, where $D_{1/2}$ is the spinor representation, and it yields the spin 1/2 basis functions $|S \uparrow\rangle$ and $|S \downarrow\rangle$. For the P-like bands one gets $T_2 \otimes D_{1/2} = \bar{\Gamma}_8 \oplus \bar{\Gamma}_7$, where $\bar{\Gamma}_8$ represents the basis functions of total angular momentum 3/2, and $\bar{\Gamma}_7$ has total angular momentum 1/2. These basis functions are listed in Table III. For GaAs and CdTe the conduction band is represented by $\bar{\Gamma}_6$ (S-type, and spin 1/2), the first valence band is composed of P-type orbitals with total angular momentum 3/2, which are described by the $\bar{\Gamma}_8$ irrep, and the split-off band contains P-type orbitals with total angular momentum 1/2, which defines the irrep $\bar{\Gamma}_7$. In contrast, for HgTe the $\bar{\Gamma}_6$ and $\bar{\Gamma}_8$ are inverted due to fine structure corrections.

The basis from Table III diagonalizes the spinful effective Hamiltonian at $\boldsymbol{k} = 0$, and leads to the well known extended Kane Hamiltonian [15]. The expression for the $8 \times 8$ Hamiltonian $H_{\text{ZB}}$ is shown in Appendix C in terms of the coefficients $c_j$ following the output of the qsymm code, so that it matches Examples in our repository. There, the notation for the powers of $\boldsymbol{k}$ follows from Ref. [15], such that it can be directly compared to the extended Kane model shown in their Appendix C. The values for the coefficients $c_j$ are also shown in Appendix C.

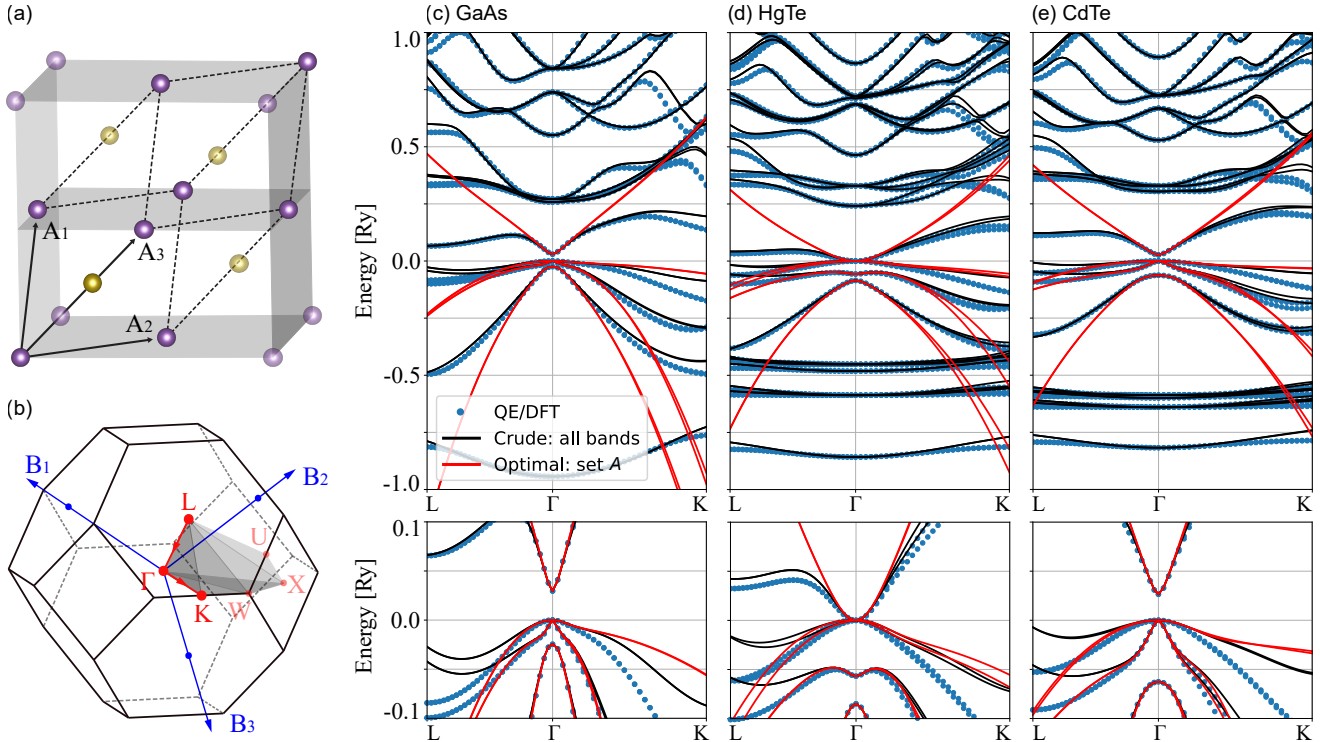

Figure 3. (a) Zincblende lattice, and (b) its first Brillouin zone (FCC). The band structure for (c) GaAs, (d) HgTe, and (e) CdTe are shown over a large energy scale on the main panels, while at the bottom of each panel, we show a zoom over the relevant low energy range. In all cases, the DFT data consider 1000 bands.

The band structures calculated from $H_{ZB}$ are shown in Fig. 3, which also shows the crystal lattice and the first Brillouin zone in Figs. 3(a-b). In all cases, Figs. 3(c–e), the blue dots represent the DFT results. The black lines are the crude model from Eq. 4, which includes all DFT bands and approaches a full zone description, but with a cost of a large $N \times N$ model with typical $N \gg 100$. More importantly, the red lines represent effective $8 \times 8$ Kane model from $H_{ZB}$, which matches well the DFT data at low energies and near $\Gamma$, as shown in the zoomed insets below each panel for GaAs [Fig. 3(c)], HgTe [Fig. 3(c)], and CdTe [Fig. 3(c)]. Particularly, for HgTe it is clear the band inversion between the $\bar{\Gamma}_6$ and $\bar{\Gamma}_8$ irreps.

Table III. Basis functions for zincblende crystals. The first column indicates the double group irreps for the $T_d$ point group at $\Gamma$, which are induced from the single group irreps in parenthesis. The second column lists the basis functions on the basis of total angular momentum, and the third column shows their expressions in terms of the symmetry orbitals (S, X, Y, Z) and spin ($\uparrow$, $\downarrow$), which follows the definitions from Ref. [15].

| IRREP $T_d$ | $\lvert J, m_j \rangle$ | $\lvert orb, spin \rangle$ |
|---|---|---|
| $\bar{\Gamma}_6(A_1)$ | $\left\lvert \frac{1}{2}, +\frac{1}{2} \right\rangle$ | $\lvert S, \uparrow \rangle$ |
| | $\left\lvert \frac{1}{2}, -\frac{1}{2} \right\rangle$ | $\lvert S, \downarrow \rangle$ |
| $\bar{\Gamma}_8(T_2)$ | $\left\lvert \frac{3}{2}, +\frac{3}{2} \right\rangle$ | $-\frac{1}{\sqrt{2}} \lvert X + iY, \uparrow \rangle$ |
| | $\left\lvert \frac{3}{2}, -\frac{3}{2} \right\rangle$ | $+\frac{1}{\sqrt{2}} \lvert X - iY, \downarrow \rangle$ |
| | $\left\lvert \frac{3}{2}, -\frac{1}{2} \right\rangle$ | $+\frac{1}{\sqrt{6}} \left[ 2 \lvert Z, \downarrow \rangle + \lvert X - iY, \uparrow \rangle \right]$ |
| | $\left\lvert \frac{3}{2}, +\frac{1}{2} \right\rangle$ | $+\frac{1}{\sqrt{6}} \left[ 2 \lvert Z, \uparrow \rangle - \lvert X + iY, \downarrow \rangle \right]$ |
| $\bar{\Gamma}_7(T_2)$ | $\left\lvert \frac{1}{2}, -\frac{1}{2} \right\rangle$ | $+\frac{1}{\sqrt{3}} \left[ \lvert Z, \downarrow \rangle - \lvert X - iY, \uparrow \rangle \right]$ |
| | $\left\lvert \frac{1}{2}, +\frac{1}{2} \right\rangle$ | $-\frac{1}{\sqrt{3}} \left[ \lvert Z, \uparrow \rangle + \lvert X + iY, \downarrow \rangle \right]$ |

## B. Wurtzite crystals

The wurtzite crystals form a lattice that is characterized by the space group P6$_3$mc, and the low energy band structure appears near the $\Gamma$ point only. Near $\Gamma$, one can factorize the translations and the resulting factor group is the $C_{6V}$ point group, which is generated by the $C_6$ rotation around the z-axis, and the mirror $M_x$. Here, in terms of the crystallographic coordinates, $x \parallel [100]$, $y \parallel [010]$, and $z \parallel [001]$. The unit cell and first Brillouin zone for these materials are shown in Figs. 4(a) and 4(b).

To illustrate the results for wurtzite materials, we consider the cases of GaN, GaP, and InP. Their band structures are shown in Figs. 4(c–e). In all cases, the top valence bands are characterized by the irreps $(A_1 + E_1) \otimes D_{1/2} = \bar{\Gamma}_7 \oplus 2\bar{\Gamma}_9$. Here, $A_1$ is the trivial irrep of $C_{6V}$ (single group), which represents S-like and Z-like orbitals, and $E_1$ is the vector representation of $C_{6V}$ that contains (X, Y)-like orbitals. These are composed with the pure spinor representation $D_{1/2}$ to define the $C_{6V}$ double group irreps $\bar{\Gamma}_7$ and $\bar{\Gamma}_9$. Additionally, we consider two conduction bands, which are characterized by the irreps $(A_1 + B_1) \otimes D_{1/2} = \bar{\Gamma}_8 \oplus \bar{\Gamma}_9$. The orbital basis function for the $B_1$ irrep is odd under both $C_6$ and $M_x$, its representation on group character tables is cumbersome, so one defines it as $\left|X(X^2 - 3Y^2)\right\rangle \equiv |V\rangle$ [14]. Ultimately, we consider the double group representations ordered as shown in Table IV.

Table IV. Basis functions for wurtzite crystals. The first column shows the double group irreps of $C_{6V}$, which are induced from the single group irrep between parenthesis. The second column shows the basis representation in terms of the spherical harmonics $Y_l^m$ and spin ($\uparrow$, $\downarrow$), while the third column shows the representation in terms of the orbitals (S, X, Y, Z, V), where $V = X(X^2 - 3Y^2)$ [14].

| IRREP $C_{6V}$ | $\|Y_l^m, \text{spin}\rangle$ | $\|\text{orb}, \text{spin}\rangle$ |
|---|---|---|
| $\bar{\Gamma}_9^c(A_1)$ | $\left\|Y_0^0, \uparrow\right\rangle$ | $\|S', \uparrow\rangle$ |
| | $\left\|Y_0^0, \downarrow\right\rangle$ | $\|S', \downarrow\rangle$ |
| $\bar{\Gamma}_8^c(B_1)$ | $\left\|Y_3^3 - Y_3^{-3}, \uparrow\right\rangle$ | $\|V, \uparrow\rangle$ |
| | $\left\|Y_3^3 - Y_3^{-3}, \downarrow\right\rangle$ | $\|V, \downarrow\rangle$ |
| $\bar{\Gamma}_9^v(A_1)$ | $\left\|Y_1^0, \uparrow\right\rangle$ | $\|Z', \uparrow\rangle$ |
| | $\left\|Y_1^0, \downarrow\right\rangle$ | $\|Z', \downarrow\rangle$ |
| $\bar{\Gamma}_9^v(E_1)$ | $\left\|Y_1^1, \uparrow\right\rangle$ | $\|X' + iY', \uparrow\rangle$ |
| | $\left\|Y_1^{-1}, \downarrow\right\rangle$ | $\|X' - iY', \downarrow\rangle$ |
| $\bar{\Gamma}_7^v(E_1)$ | $\left\|Y_1^{-1}, \uparrow\right\rangle$ | $\|X' - iY', \uparrow\rangle$ |
| | $\left\|Y_1^1, \downarrow\right\rangle$ | $\|X' + iY', \downarrow\rangle$ |

There the top indexes $\{c, v\}$ refer to conduction and valence bands. Notice that the $\Gamma_9$ irrep appears in three pairs of basis functions, which allows for the $s$–$p_z$ mixing [110–112] Here, however, we always work on the diagonal basis ($H_{WZ}$ is diagonal at $\boldsymbol{k} = 0$), which is indicated by the primes in the orbitals above. For a recent and detailed discussion on this choice of representation and the $s$–$p_z$ mixing, please refer to Ref. [113].

Using the basis functions from Table IV to calculate the effective $10 \times 10$ model using qsymm, we obtain the Hamiltonian $H_{WZ}$ shown in Appendix C. Here we always consider two conduction bands, which leads to this $10 \times 10$ generic model $H_{WZ}$. However, one can also opt to work with traditional $8 \times 8$ models with a single conduction band. Notice, however, that for GaP the first conduction band transforms as $\bar{\Gamma}_8$, while for GaN and InP the first conduction band is $\bar{\Gamma}_9$. Therefore, one must be careful when selecting the appropriate $8 \times 8$ model for wurtzite materials. For the valence bands, one always gets $\bar{\Gamma}_7 \oplus 2\bar{\Gamma}_9$, however, the internal ordering of these valence bands may change between materials and it can be highly sensible to the choice of density functional [28, 60, 114, 115]. The numerical coefficients $c_j$ found for GaN, GaP, InP are shown in Appendix C, and the resulting band structures are shown in Figs. 4(c–e). In all cases, we see that the crude model with 1000 bands (black lines) approaches a full zone description, but here we are more interested in the reduced $10 \times 10$ models (red lines), which present satisfactory agreement with the DFT data at low energies.

### 1. Effects of the SOC corrections on $\boldsymbol{P}_{m,n}$

As introduced in Sec. II D 2, the matrix elements $\boldsymbol{P}_{m,n}$ can be calculated with or without the PAW corrections, $\boldsymbol{p}_{SOC}$, that carry the SOC contributions. For most of the materials we have studied here, these corrections are marginal and the results from both cases are nearly identical. Nevertheless, we emphasize that using our patched bands.x within QE is faster than using the Python code to calculate $\boldsymbol{P}_{m,n}$ via Eq. (13).

To illustrate the effects of the PAW/SOC corrections on the matrix elements $\boldsymbol{P}_{m,n}$, Fig. 5 compares the models for GaN and GaP with and without these corrections. For the conduction bands, we notice that the $\boldsymbol{p}_{SOC}$ corrections significantly improve the GaN effective mass, but barely affect GaP. For the valence bands, both GaN and GaP show moderate effects of $\boldsymbol{p}_{SOC}$. Indeed, this shows that a precise calculation of $\boldsymbol{P}_{m,n}$ is critical to improve the precision of the models [116].

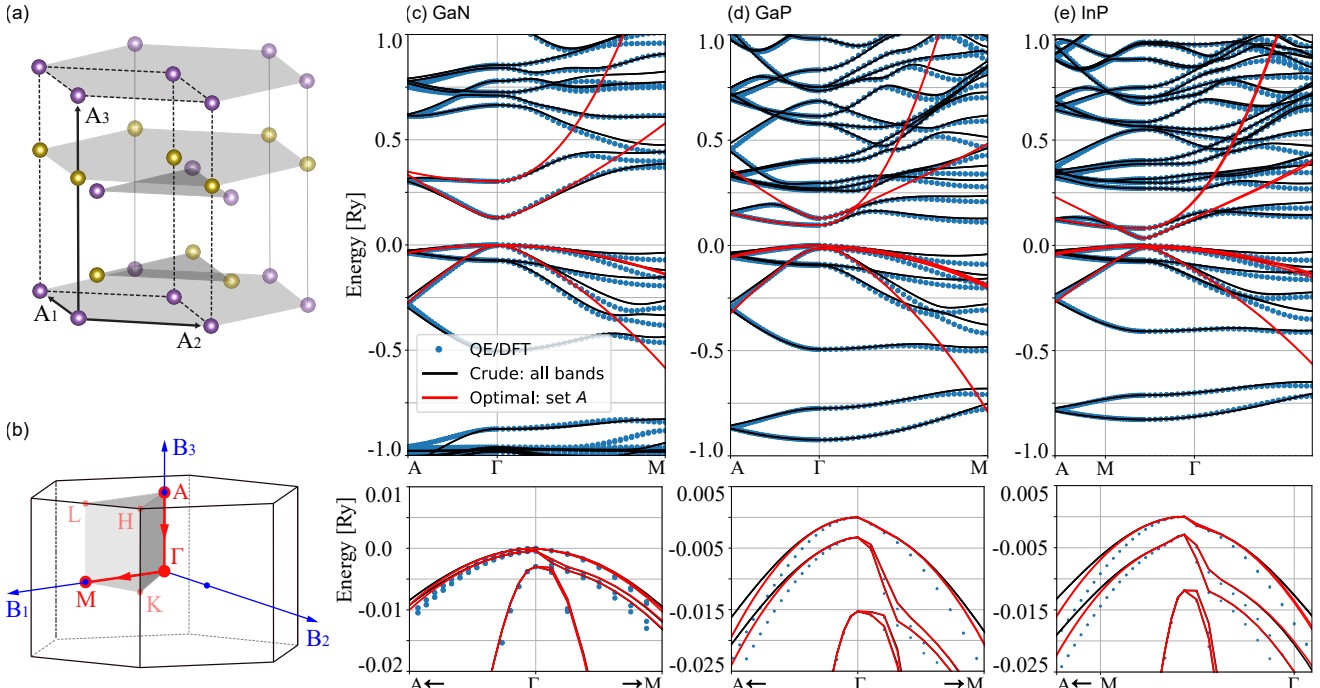

Figure 4. (a) Lattice and (b) Brillouin zone for wurtzite crystals. Band structures for (c) GaN, (d) GaP, and (e) InP show the large energy range on top, and a zoom shows the top of the valence bands at the bottom of each panel. In all cases, the DFT calculation considers 1000 bands.

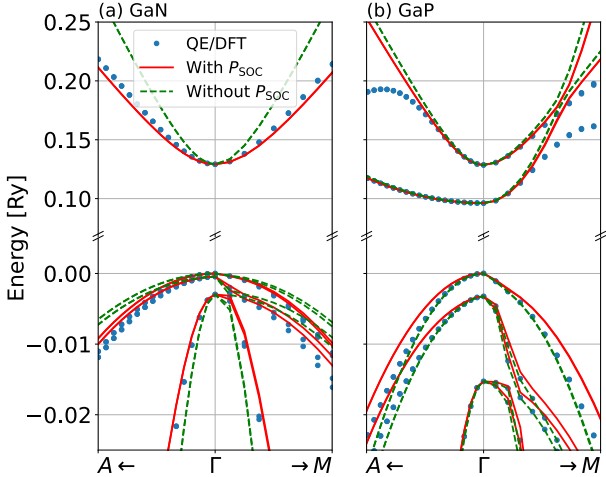

Figure 5. Comparison between the DFT data and the effective models calculated with the full matrix element $\boldsymbol{P}_{m,n}$ including PAW/SOC corrections (red lines) and the simplified $\boldsymbol{P}_{m,n}$ without PAW/SOC corrections (green lines) for (a) GaN and (b) GaP.

## C.  Rock-salt crystals

The crystal lattice for rock-salt crystals is shown in Fig. 6(a), which is an FCC lattice with two atoms in the base, and it is described by the space group Fm$\bar{3}$m. The

low energy band structure concentrates at the L point of the Brillouin zone shown in Fig. 6(b), which transforms as the $D_{3D}$ point group after factorizing the Bloch translations. The basis functions for the first valence and conduction bands transform as $A_{1g} \otimes D_{1/2} = \bar{L}_6^+$ and $A_{2u} \otimes D_{1/2} = \bar{L}_6^-$, where $A_{1g}$ is the trivial irrep for S-like orbitals, and $A_{2u}$ represent Z-like orbitals [117]. Therefore, the basis functions for the $\bar{L}_6^+$ bands are $\{|S,\uparrow\rangle, |S,\downarrow\rangle\}$, and for $\bar{L}_6^-$ one gets $\{|Z,\uparrow\rangle, |Z,\downarrow\rangle\}$. Here, the $x$, $y$, and $z$ coordinates are taken along the $[\bar{1}\bar{1}2]$, $[1\bar{1}0]$, and $[111]$ crystallographic directions.

Here we consider two examples of rock-salt crystals: PbSe and SnTe. Their effective $4 \times 4$ Hamiltonian $H_{\mathrm{RS}}$ under the $\bar{L}_6^{\pm}$ basis, and its numerical parameters are shown in Appendix C, and the comparison between DFT and model band structures are shown in Figs. 6(c)–(d). PbSe is a narrow gap semiconductor, where the conduction band transforms as the $\bar{L}_6^+$ irrep, and the valence band as $\bar{L}_6^-$. In contrast, SnTe shows inverted bands, with $\bar{L}_6^+$ below $\bar{L}_6^-$, yielding a topological insulator phase [118, 119]. In both cases, the low-energy model captures the main features of the bands, including the anisotropy.

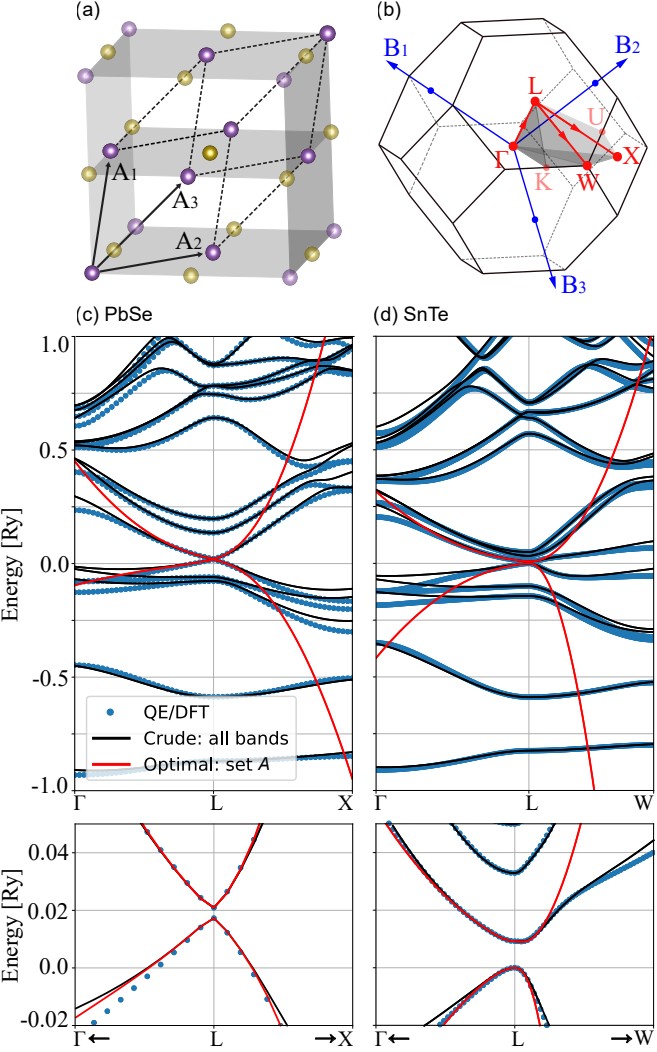

Figure 6. (a) The rock salt lattice and (b) its Brillouin zone (FCC). Band structures for (c) PbSe and (d) SnTe. The bottom of each panel zooms into the low energy range near the Fermi level. Both DFT calculations were performed considering 500 bands.

Table V. Summary of space group, irreps and basis functions for the low energy bands of $MoS_2$, $GaBiCl_2$, and $Bi_2Se_3$. The first column lists the materials, the second indicates the lattice space group, and the little group at the relevant k point. The third and fourth columns lists the irreps and basis functions for the low energy bands in each case. The table shows the double group irreps and the corresponding single group irreps between parenthesis.

| Material | Group info | IRREP | Basis |
|---|---|---|---|
| $MoS_2$ | Space group | $\bar{K}_{11}(E_1')$ | $\lvert X+iY,\uparrow\rangle$ |
| | $P\bar{6}m2$ | $\bar{K}_{10}(E_1')$ | $\lvert X+iY,\downarrow\rangle$ |
| | Little group | $\bar{K}_8(A')$ | $\lvert S,\uparrow\rangle$ |
| | K: $C_{3h}$ | $\bar{K}_9(A')$ | $\lvert S,\downarrow\rangle$ |
| $GaBiCl_2$ | Space group | $\bar{\Gamma}_4(E)$ | $\lvert X+iY,\uparrow\rangle$ |
| | | $\bar{\Gamma}_5(E)$ | $\lvert X-iY,\downarrow\rangle$ |
| | P3m1 | $\bar{\Gamma}_6(E)$ | $\lvert X-iY,\uparrow\rangle$ |
| | | | $\lvert X+iY,\downarrow\rangle$ |
| | Little group | $\bar{\Gamma}_6(A_1)$ | $\lvert Z\uparrow\rangle$ |
| | $\Gamma$: $C_{3V}$ | | $\lvert Z\downarrow\rangle$ |
| $Bi_2Se_3$ | Space group | $\bar{\Gamma}_6^+(A_{1g})$ | $\lvert S,\uparrow\rangle$ |
| | $R\bar{3}m$ | | $\lvert S,\downarrow\rangle$ |
| | Little group | $\bar{\Gamma}_6^-(A_{2u})$ | $\lvert Z,\uparrow\rangle$ |
| | $\Gamma$: $D_{3d}$ | | $\lvert Z,\downarrow\rangle$ |

### D. Other examples

To finish the set of illustrative examples, we show here the case for: (i) the monolayer $MoS_2$, which is one of the most studied transition metal dichalcogenides (TMDC) [120–122]; (ii) the bulk bismuth selenide ($Bi_2Se_3$), which is one of the first discovered 3D topological insulators [123, 124]; and (iii) a monolayer of $GaBiCl_2$, which is a large gap 2D topological insulator [125]. The symmetry characteristics and basis functions for the low-energy bands of these materials mentioned above are summarized in Table V.

For $MoS_2$, the first valence and conduction bands are given by the single group irreps $A'$ and $E_1'$ of the $C_{3h}$ group [39, 126], which can be represented as S-like and $(X+iY)$-like orbitals. For $GaBiCl_2$, the valence bands are characterized by single group $E$ irrep, and it splits into $E\otimes D_{1/2}=\bar{\Gamma}_4\oplus\bar{\Gamma}_5\oplus\bar{\Gamma}_6$ in the spinful case, while the conduction band is given by the irrep $A_1\otimes D_{1/2}=\bar{\Gamma}_6$. For $Bi_2Se_3$, a detailed derivation of the effective model can be seen in Ref. [127], which shows that the first valence and conduction bands are given by $A_{1g}\otimes D_{1/2}=\Gamma_6^+$, and $A_{2u}\otimes D_{1/2}=\Gamma_6^-$.

The effective Hamiltonians and their numerical coefficients for these materials can be found in the `Examples` folder of the code repository. Here we show only the comparison between the DFT and model band structures in Fig. 7. The $MoS_2$ case, as shown in Fig. 7(a), is challenging for a $\boldsymbol{k}\cdot\boldsymbol{p}$ method, since its band structure presents valleys in between high symmetry points. Consequently, the 4 bands model (red lines) captures only the nearly parabolic dispersion at the K point.

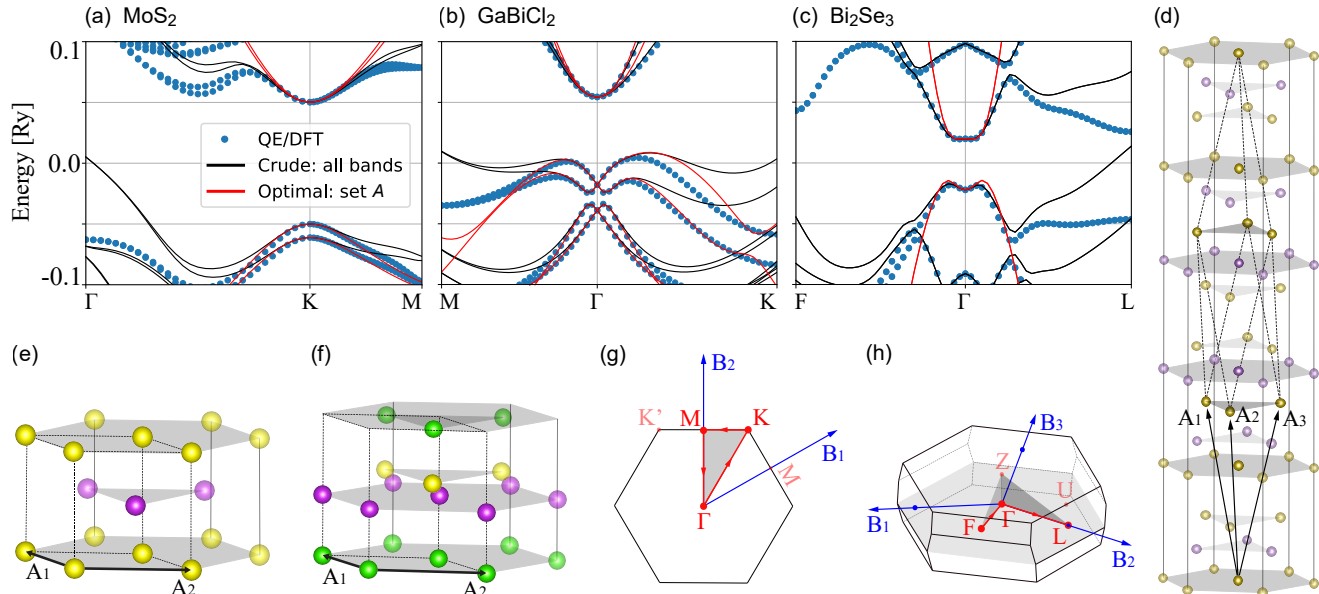

Figure 7. Band structures for: (a) MoS$_2$, (b) GaBiCl$_2$, and (c) Bi$_2$Se$_3$ showing only the relevant low energy range. The DFT calculations were performed for 1000, 500, and 500 bands, respectively. (d) Rhombohedral lattice of Bi$_2$Se$_3$ and 2D hexagonal lattice of (e) MoS$_2$ and (f) GaBiCl$_2$, where we have omitted the vacuum region (15 Å) perpendicular to the plane formed by vectors A$_1$ and A$_2$. (g) 2D Brillouin zone common to MoS$_2$ and GaBiCl$_2$, and (h) 3D BZ of Bi$_2$Se$_3$.

However, the crude all-bands model (black lines, see Eq. (4)) approaches a full zone description and captures the valley along the Γ–K direction. For GaBiCl$_2$, Fig. 7(c), the 6 bands model describes satisfactorily the low energy conduction and valence bands. For Bi$_2$Se$_3$ in Fig. 7(b) the 4 bands model captures well the low-energy band structure near Γ, including the hybridization between the inverted bands.

## V.  DISCUSSIONS

Above, we have presented illustrative results of the capabilities of our code to calculate the $\boldsymbol{k} \cdot \boldsymbol{p}$ Kane and Luttinger parameters for a series of relevant materials. In all cases we see a patent agreement between the DFT (QE) data and the low-energy models near the relevant $\boldsymbol{k}_0$ point. However, it is important to notice that here we use only PBE functionals [128], consequently it often underestimates the gap (e.g. 0.5 eV instead of 1.5 eV for GaAs). Therefore, our models are limited by the quality of the DFT bands and the resulting numerical parameters might not match Kane and Luttinger's parameters for well-known materials, for which these parameters are typically chosen to match the experimental data, and not the DFT simulations.

For instance, let us consider the zincblende crystals' Kane parameter $E_P = 2m_0 P^2/\hbar^2$, band gap $E_g$ and effective mass for the conduction band $m^*$. For GaAs, the experimental values are $E_P \sim 24$ eV, $P \sim 0.96$ eVnm, $E_g \sim 1.5$ eV, and $m^* = 0.065m_0$ [58]. As mentioned above, the DFT results with PBE functionals underes-

timate the gap, and we get $E_g \sim 0.5$ eV. Moreover, the Kane parameter can be written as $P = -\sqrt{6}c_5/2$, where the coefficient $c_5 = -0.635$ eVnm is shown in Appendix C. This value yields $P \sim 0.7$ eVnm and $E_P \sim 16$ eV. The effective mass for the conduction band can be estimated from its spinless expression [47], $m_0/m^* = 1 + 2m_0 P^2/E_g \hbar^2$, which gives us $m^* = 0.031m_0$. While these numbers do not match well with the experimental values, we notice that if we fix the GaAs gap (scissors-cut approximation), but keep our value for $P$, we find $m^* = 0.058m_0$, which is already much closer to the experimental value for the effective mass.

The number estimates shown above clearly indicate that the quality of our models is limited to the DFT simulations only. Particularly, the gap issue can be fixed if one replaces the PBE functionals with hybrid functionals, GW calculations, or other methods that improve the material gap accuracy. These are beyond the scope of this paper, but it is a possible path for future improvements of our code.

In all examples presented here, we always consider the crude all bands model from Eq. (4), and the optimal symmetry-adapted (few bands) model from Eq. (5). This raises two interesting questions: (i) how many bands are necessary for convergence? And (ii) for a large number of bands, should we get a full zone description? We discuss these questions below.

## A. Convergence

The convergence threshold (how many bands are necessary) strongly depends on the material. In some cases $\sim 300$ bands are sufficient, but in others, it often needs $\sim 1000$ bands. We do not have a general rule to establish which materials will show a slow or fast convergence. Nevertheless, we believe it is instructive to discuss the outcomes of our convergence analysis.

Notice that the Löwdin partitioning from Eq. (5) has two distinct contributions. The first two terms in Eq. (5) are the zeroth and first-order perturbation terms. These terms do not change as we increase the number of DFT bands (provided that there are enough bands to converge the DFT calculation itself). The zeroth order term is essentially given by the DFT eigenstates, and the first order terms are given by the matrix elements $\langle m| H'(\boldsymbol{k}) |n\rangle = 2\boldsymbol{k} \cdot \boldsymbol{P}_{m,n}$ between eigenstates of set $A$, which is the low energy sector of interest. In contrast, the third term defines the second-order corrections, which are quadratic in $\boldsymbol{k}$ (assuming a diagonal basis at $\boldsymbol{k} = 0$). In this case, the second-order contributions depend explicitly on the sum over the remote set of bands $B$. These are the terms that strongly depend on the number of remote bands.

To check for convergence, we plot the values of the Hamiltonian coefficients $c_j$ associated with second-order corrections as a function of the number of remote bands. In the `Examples` folder in the code repository, one finds these plots for all cases presented in this paper. Here, in the top panels of Fig. 8, we select a few illustrative cases. In the bottom panels of Fig. 8 we combine the discrete derivatives of $c_j$ into a single dimensionless metric for convergence $\mathcal{C}(N)$, which read as

$$\mathcal{C}(N) = \frac{\sum_j |c_j(N+1) - c_j(N)|}{\sum_j |c_j(N)|}, \qquad (35)$$

where $c_j(N)$ refers to the coefficient calculated using $N$ remote bands. With increasing $N$, the coefficients are expected to converge, consequently $\mathcal{C}(N) \to 0$. The data for $\mathcal{C}(N)$ is shown in blue dots on the bottom panels of Fig. 8, which is significantly noisy due to the discrete jumps on the evolution of $c_j$ with increasing $N$. Therefore, we also plot a moving average $\mathcal{C}(N)$ (orange lines) to clearly show the convergence. For spinless graphene in Fig. 8(a), there are only two second order $c_j$ terms (neglecting terms with $k_z$, since it is a 2D material), and we see that it reaches convergence with less than 300 remote bands.

In contrast, for $MoS_2$, the convergence requires at least $\sim 500$ remote bands. Interestingly, it has been recently shown that TMDC materials indeed require a large number of bands to converge the orbital angular momenta [42–45]. This fact may be associated with the large number of unoccupied bands with plane-wave character that appear due to the spatial extension of the vacuum region. The GaN and GaP cases in Figs. 8(c)–(d) are interesting cases, they belong to the same class of materials,

but GaP reaches convergence with $\sim 200$ remote bands, while GaN is not yet fully converged for $\sim 1000$ remote bands. Unlike monolayer materials, the GaN compound is not described by any vacuum region, and therefore we speculate that such poor convergence may be related to details of the pseudopotential [100] and the electronegativity of Nitrogen.

## B. Full zone kp

In Section II A we have presented the $\boldsymbol{k} \cdot \boldsymbol{p}$ method in its traditional form, which considers a perturbative expansion of the Bloch Hamiltonian at a reference momentum $\boldsymbol{k}_0$, and a small set of bands near the Fermi energy. Usually, one expects the resulting effective model to be valid only near $\boldsymbol{k}_0$ and only for a small energy range that encloses the bands of interest. In contrast, within the *full zone* $\boldsymbol{k} \cdot \boldsymbol{p}$ approach [66, 129–133] one considers a large set of bands, such that the resulting low energy model agrees well with DFT or experimental bands over the full Brillouin zone, instead of only the vicinity of $\boldsymbol{k}_0$. However, to achieve this precision, one needs to apply fitting procedures to ensure that the bands match selected energy levels at various $\boldsymbol{k}$ points over the Brillouin zone.

Here, in our code, we can easily select an arbitrary number of bands to build effective models. All examples presented above show sets of bands colored in red and black, such that the red ones consider models built from a small set of bands $A$ (from 4 to 10 bands), while the black ones consider the full set of bands from the DFT data (typically 500 or 1000 bands). This leads to an interesting question: should our *all bands model* match the *full zone* $\boldsymbol{k} \cdot \boldsymbol{p}$ models?

To answer this question, let us focus first on the graphene results from Fig. 1. There, we have seen that the QE/DFT and the model agree remarkably well at low energies near the K point, as expected. Particularly, the red line for the optimal symmetry-adapted model describes precisely the low energy regime and Dirac cone and the trigonal warping from the quadratic terms in Eq. (21). In contrast, when we consider the all-bands model (black lines), we see that the model approaches a full zone agreement with 300 bands. What if we consider more bands? Our numerical tests have shown that increasing the number of bands does improve the overall description, approaching the full zone agreement. However, this is a very slow convergence and we never really reach a true full zone agreement. This characteristic is seen in all other examples shown here.

For GaAs, Gawarecki and collaborators [133] show an excellent full zone agreement between model and DFT bands considering 30 bands. In contrast, our results presented in Fig. 3(a) for 8 (red) and 1000 (black) bands remain valid only in the vicinity of $\Gamma$. The key difference is the fitting procedure. The full zone models fit the bands over the full Brillouin zone, while in our approach we consider only the direct *ab initio* matrix elements of

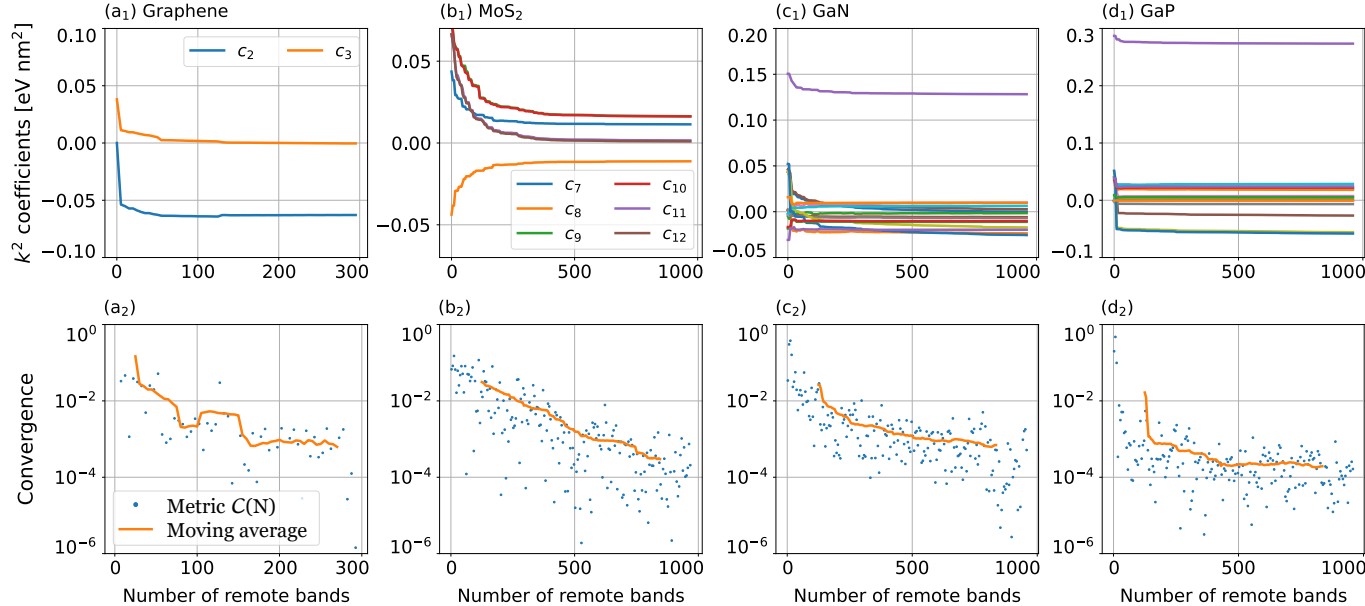

Figure 8. Convergence of the second-order coefficients $c_j$ as a function of the number of remote bands for (a) spinless graphene, (b) $MoS_2$, (c) GaN, and (d) GaP. On top ($a_1$–$d_1$), each panel shows the coefficients $c_j$ for different material. On panels ($c_1$) and ($d_1$) we omit the legends because there are 30 distinct coefficients, ranging from $c_{22}$ to $c_{51}$, which makes their individual identification cumbersome, and it is sufficient to visualize that all lines become nearly flat for a large number of remote bands. On the bottom ($a_2$–$d_2$), for each material, the evolution of the coefficients $c_j$ are combined into convergence metric set by Eq. (35) (blue dots). Due to the noise induced by the discrete derivative in this metric, we plot the moving average of the data as a guide for the eyes.

$\boldsymbol{\pi} = \boldsymbol{p} + \boldsymbol{p}_{\mathrm{SOC}}$ without further manipulation.

If one needs a full zone model, we suggest using our results as the initial guess for the parameters used on a band-fitting algorithm. Moreover, since the fitted parameters must not deviate significantly from our *ab initio* results, our calculated values provide an important benchmark for the fitting results. Alternatively, it might be possible to develop multi-valley $\boldsymbol{k} \cdot \boldsymbol{p}$ models [68, 70, 134] and extract its parameters directly from DFT matrix elements without numerical fitting procedures, but this is beyond the scope of this work.

## VI. CONCLUSIONS

We have implemented a numerical framework to calculate the $\boldsymbol{k} \cdot \boldsymbol{p}$ Kane and Luttinger parameters and optimal symmetry-adapted effective Hamiltonians directly from *ab initio* wavefunctions. The code is mostly written in Python but also contains a patch to modify the Quantum ESPRESSO code, such that its `bands.x` post processing tool is used to calculate the matrix elements $\boldsymbol{P}_{m,n} = \langle m | \boldsymbol{\pi} | n \rangle$, which is the central quantity in our methodology. Consequently, this first version works only with Quantum ESPRESSO. Equivalent calculations can be done in other DFT codes (e.g. VASP [5], Wien2k [6]), but it requires further developments. The code is open source and it is available at Ref. [89].

Here, we have illustrated the capabilities of our code

applying it to a series of relevant and well-known materials. The resulting effective models yield band structures that match well the DFT data in the low energy sector near the k point used for the wavefunction expansion. Therefore, our code provides an *ab initio* approach for the $\boldsymbol{k} \cdot \boldsymbol{p}$ numerical parameters, which can be contrasted with fitting methods [60, 76–78, 133], in which the numerical coefficients are obtained by numerically minimizing the residue difference between the DFT and model band structures over a selected range of the Brillouin zone. These fitting procedures work well in general but require careful verification if the fitted parameters are reasonable. In contrast, our *ab initio* approach is automatic and fully reliable. Nevertheless, fitting procedures can improve the agreement between DFT and the model band structures significantly. In this case, we suggest that our code can be used (i) to generate the initial values for the fitting parameters, and (ii) to verify if the fitted parameters show reasonable values. One should expect that fitted parameters must not deviate much from our *ab initio* values.

Here we do not perform a thorough comparison of our numerical parameters with experimental data. Typically, to obtain precise agreement with experimental data, one needs to fix the gap issue by using either hybrid functionals or GW calculations, which are beyond the scope of this first version of the code. Instead, here we use only PBE functionals [128] for simplicity, which is reliable enough to validate our approach. Consequently, our

numerical parameters are limited by the precision of the DFT simulation, and we would not expect remarkable agreement with experimental data for most materials at this stage. Nevertheless, for novel materials, for which there is no experimental data available, our code can be used to generate reliable numerical parameters that can be improved later, either in comparison with future experiments or by extending our method to work with hybrid functionals or GW calculations.

As a final disclaimer, we would like to state that after developing the first version of the code, we have found that Ref. [86] recently proposes an equivalent approach to build $\boldsymbol{k} \cdot \boldsymbol{p}$ models from DFT, but the authors do not provide an open-source code. In any case, despite the similarities, the development of our code was done independently from their proposal. In practice, the only significant difference between the proposals is the approach to calculate the transformation matrix $U$ (see Section II C). While the authors of Ref. [86] follow the method from [95], here we propose a different method that is more efficient for transformations involving reducible represen-

tations, which is necessary when dealing with nearly degenerate bands of different irreps (e.g., spinful graphene). Additionally, after the initial submission of our paper, a new code VASP2kp [135] was released with functionalities similar to ours, but designed for VASP [5] instead of QE.

## ACKNOWLEDGMENTS

This work was supported by the funding agencies CNPq, CAPES, and FAPEMIG. G.J.F. acknowledges funding from the FAPEMIG grant PPM-00798-18). P.E.F.J. acknowledges the financial support of the DFG SFB 1277 (Project-ID 314695032, projects B07 and B11) and SPP 2244 (Project No. 443416183). A.L.A. acknowledges the financial support from FAPESP (grants 2022/08478-6 and 2023/12336-5). GFJ acknowledges useful discussions with P. Giannozzi about PAW parameters on QE's pseudopotentials; H. Zhao for useful discussions and the suggestion to use the IrRep and qeirreps [136] packages, and S. S. Tsirkin for discussions about the implementation of the IrRep package [92].

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

## Appendix A: Mass-velocity corrections are negligible

Consider the full Hamiltonian with all fine structure corrections as

$$H = p^2 + V(\boldsymbol{r}) + H_{\text{MV}} + H_{\text{D}} + H_{\text{SOC}}, \qquad \text{(A1)}$$

$$H_{\text{MV}} = -\frac{\alpha^2 p^4}{4}, \qquad \text{(A2)}$$

$$H_{\text{D}} = \frac{\alpha^2}{8}\nabla^2 V(\boldsymbol{r}), \qquad \text{(A3)}$$

$$H_{\text{SOC}} = \frac{\alpha^2}{4}[\boldsymbol{\sigma} \times \nabla V(\boldsymbol{r})] \cdot \boldsymbol{p}. \qquad \text{(A4)}$$

Applying the Bloch theorem $\psi_{\boldsymbol{\kappa}}(\boldsymbol{r}) = e^{i\boldsymbol{k}\cdot\boldsymbol{r}}\phi_{\boldsymbol{k}_0,\boldsymbol{k}}(\boldsymbol{r})$ for $\boldsymbol{\kappa} = \boldsymbol{k}_0 + \boldsymbol{k}$, the $\boldsymbol{k}\cdot\boldsymbol{p}$ Hamiltonian becomes $H_{\text{kp}} = H_0 + k^2 + H'$, where $H_0 = p^2 + V(\boldsymbol{r}) + 2\boldsymbol{k}_0\cdot\boldsymbol{\pi} + H_{\text{SR}}$, and $H_{\text{SR}}$ contain the $\boldsymbol{k} = 0$ contributions from $H_{\text{MV}} + H_{\text{D}}$, as presented in the main text. The perturbation for finite $\boldsymbol{k} \neq 0$ is $H' = 2\boldsymbol{k}\cdot\boldsymbol{\pi} + H'_{\text{MV}}$, where $H'_{\text{MV}}$ contains the finite $\boldsymbol{k}$ contributions from the mass velocity term, and it reads as

$$H'_{\text{MV}} = -\frac{\alpha^2}{4}\Big[4(\boldsymbol{k}\cdot\boldsymbol{p})p^2 + 4(\boldsymbol{k}\cdot\boldsymbol{p})^2$$
$$+ 4(k)^2(\boldsymbol{k}\cdot\boldsymbol{p}) + 2k^2 p^2 + k^4\Big]. \quad \text{(A5)}$$

These corrections are negligible for small $\boldsymbol{k}$, i.e. $|H'_{\text{MV}}| \ll |2\boldsymbol{k}\cdot\boldsymbol{\pi}|$. Notice that the SOC term in $2\boldsymbol{k}\cdot\boldsymbol{\pi}$ has two contributions, one is of order $\sim |kp|$ and the other is $\sim |k\alpha^2|$. In contrast, the contributions to $H'_{\text{MV}}$ are $\sim |\alpha^2 kp^3|$, $\sim |\alpha^2 k^2 p^2|$, $\sim |\alpha^2 k^3 p|$, and $\sim |\alpha^2 k^4|$. Therefore, all terms in $H'_{\text{MV}}$ are of higher order than those in $2\boldsymbol{k}\cdot\boldsymbol{\pi}$, and we can safely assume $H' \approx 2\boldsymbol{k}\cdot\boldsymbol{\pi}$.

## Appendix B: DFT parameters

The first principles calculations are performed using the density functional theory (DFT) [1, 2] within the generalized gradient approximation (GGA) for the exchange and correlation functional, employing the Perdew-Burke-Ernzerhof (PBE) parametrization [128]. We employ the non-colinear spin-DFT formalism self-consistently with fully relativistic j-dependent ONCV (Optimized Norm-Conserving Vanderbilt) pseudopotential [99]. The Quantum ESPRESSO (QE) package [3, 4] was used, with a plane waves base configured with a given cut-off energy and the Brillouin zone sampled with several k-points (Monkhorst-Pack grid) so that the total energy converged within the meV scale (see Table VI). The ONCV pseudopotentials compatible with the Quantum ESPRESSO package are available in the repository [109]. The vacuum space in two-dimensional materials was set to 15 Å. Atomic structures were optimized with a criterion that requires the force on each atom to be less than 0.01 eV/Å. Additional parameters used in our simulations including QE input and output files can be found in the Examples folder of the code repository [89].

Table VI. Criteria used for the convergence of the total energy: cut-off energy for the expansion in plane waves and the number of k-points taken for sampling the Brillouin zone using the Monkhorst-Pack technique.

| Material | cut-off energy | BZ sample |
|---|---|---|
| Graphene | 80 Ry | 12x12x1 |
| GaAs | 100 Ry | 8x8x8 |
| HgTe | 50 Ry | 8x8x8 |
| CdTe | 60 Ry | 8x8x8 |
| GaN | 100 Ry | 8x8x8 |
| GaP | 150 Ry | 8x8x8 |
| InP | 100 Ry | 7x7x7 |
| PbSe | 100 Ry | 7x7x7 |
| SnTe | 100 Ry | 8x8x8 |
| $MoS_2$ | 100 Ry | 8x8x1 |
| $Bi_2Se_3$ | 60 Ry | 7x7x7 |
| $GaBiCl_2$ | 100 Ry | 8x8x1 |

## Appendix C: Effective Hamiltonians and coefficients

Here we present the large Hamiltonians and table of parameters for the materials presented in the main text. These correspond to the zincblende crystals for Fig. 3, wurtzite crystals of Fig. 4, and rock-salt crystals of Fig. 6. For the other examples shown in Fig. 7, the corresponding Hamiltonians and numerical parameters can be seen in Examples folder in the code repository.

Table VII. Table of parameters for the zincblende materials, where the coefficients $c_n$ refer to the terms of $H_{\text{ZB}}$ in the equation listed in Table X. The coefficient $c_0$ is negative for HgTe due to the $\Gamma_6$–$\Gamma_8$ band inversion.

| Zincblende | GaAs | HgTe | CdTe |
|---|---|---|---|
| $c_0$ (eV) | 0.403 | -1.16 | 0.36 |
| $c_1$ (eV) | 0.00011 | 2.23e-05 | 3.68e-05 |
| $c_2$ (eV) | -0.335 | -0.773 | -0.851 |
| $c_3$ (eV nm) | 0.000486 | -0.0117 | 0.00232 |
| $c_4$ (eV nm) | 0.00268 | -0.023 | 0.00499 |
| $c_5$ (eV nm) | -0.635 | -0.543 | 0.559 |
| $c_6$ (eV nm) | -0.436 | 0.341 | 0.363 |
| $c_7$ (eV nm$^2$) | 0.0293 | 0.0354 | 0.0347 |
| $c_8$ (eV nm$^2$) | -0.0978 | -0.0772 | -0.0577 |
| $c_9$ (eV nm$^2$) | -0.0437 | -0.0339 | -0.0262 |
| $c_{10}$ (eV nm$^2$) | -0.0321 | 0.0128 | -0.0153 |
| $c_{11}$ (eV nm$^2$) | -0.0608 | -0.0375 | -0.0303 |
| $c_{12}$ (eV nm$^2$) | -0.000588 | -0.0036 | -0.000109 |
| $c_{13}$ (eV nm$^2$) | 0.0632 | 0.0558 | 0.0398 |
| $c_{14}$ (eV nm$^2$) | 0.0397 | -0.0259 | 0.0231 |
| $c_{15}$ (eV nm$^2$) | -0.0362 | 0.0479 | 0.0361 |
| $c_{16}$ (eV nm$^2$) | -0.0275 | -0.0349 | 0.0261 |

The numerical coefficients for the zincblende, wurtzite, and rock-salt materials are shown in Tables VII, IX, and VIII, respectively. These correspond to the effective Hamiltonians shown in Tables X, XI, and XII. In all cases we use $k_\pm = k_x \pm i k_y$, $k^2 = k_x^2 + k_y^2 + k_z^2$, $k_\parallel^2 = k_x^2 + k_y^2$, $\hat{K} = k_x^2 - k_y^2$, which is also used in Appendix C of Ref. [15].

Table VIII. Table of parameters for the rock-salt materials, where the coefficients $c_n$ refer to the terms of $H_{\mathrm{RS}}$ in the equation listed in Table XII.

| Rock-salt | PbSe | SnTe |
|---|---|---|
| $c_0$ (eV) | 0.235 | 0.125 |
| $c_1$ (eV) | 0.284 | 0.000141 |
| $c_2$ (eV nm) | 0.168 | 0.193 |
| $c_3$ (eV nm) | -0.122 | -0.111 |
| $c_4$ (eV nm$^2$) | -0.134 | -0.713 |
| $c_5$ (eV nm$^2$) | 0.223 | 0.214 |
| $c_6$ (eV nm$^2$) | 0.119 | 0.637 |
| $c_7$ (eV nm$^2$) | -0.151 | -0.158 |

Table IX. Table of parameters for the wurtzite materials, where the coefficients $c_n$ refer to the terms of $H_{\mathrm{WZ}}$ in the equation listed in Table XI.

| Wurtzite | GaP | GaN | InP |
|---|---|---|---|
| $c_0$ (eV) | 1.75 | 1.76 | 0.457 |
| $c_1$ (eV) | 9.73e-06 | -1.16e-07 | 1.4e-07 |
| $c_2$ (eV) | -6.28e-06 | -5.09e-09 | -4.07e-06 |
| $c_3$ (eV) | 1.31 | 4.11 | 1.1 |
| $c_4$ (eV) | -0.208 | -0.0405 | -0.162 |
| $c_5$ (eV) | 4.7e-08 | 0.000658 | 6.89e-09 |
| $c_6$ (eV) | -0.0442 | -0.00602 | -0.0395 |
| $c_7$ (eV) | 7.82e-05 | -7.29e-05 | 8.11e-05 |
| $c_8$ (eV nm) | 0.00448 | 0.00586 | -0.0112 |
| $c_9$ (eV nm) | 0.00214 | 0.00075 | 0.0137 |
| $c_{10}$ (eV nm) | 0.118 | -0.0733 | -0.184 |
| $c_{11}$ (eV nm) | 0.455 | -0.372 | -0.392 |
| $c_{12}$ (eV nm) | -0.472 | -0.381 | 0.436 |
| $c_{13}$ (eV nm) | -0.00429 | -0.00428 | -0.0195 |
| $c_{14}$ (eV nm) | 0.00811 | 0.0024 | 0.0223 |
| $c_{15}$ (eV nm) | 0.0234 | 0.0128 | 0.0301 |
| $c_{16}$ (eV nm) | -0.0268 | 0.0134 | -0.0428 |
| $c_{17}$ (eV nm) | -0.0112 | -0.00416 | -0.0377 |
| $c_{18}$ (eV nm) | 0.0055 | -0.00109 | 0.0202 |
| $c_{19}$ (eV nm) | 0.801 | -0.568 | -0.616 |
| $c_{20}$ (eV nm) | 0.214 | -0.116 | -0.298 |
| $c_{21}$ (eV nm) | -0.00918 | -0.00423 | -0.0294 |
| $c_{22}$ (eV nm$^2$) | 0.0203 | 0.0266 | 0.0282 |
| $c_{23}$ (eV nm$^2$) | 0.0182 | 0.00155 | -0.0109 |
| $c_{24}$ (eV nm$^2$) | 0.00486 | 0.000225 | -0.00585 |
| $c_{25}$ (eV nm$^2$) | -2.32e-05 | 8.08e-05 | -0.000406 |
| $c_{26}$ (eV nm$^2$) | 0.273 | 0.128 | 0.264 |
| $c_{27}$ (eV nm$^2$) | -0.0267 | -0.0151 | -0.0262 |
| $c_{28}$ (eV nm$^2$) | 0.00735 | 0.00235 | 0.00881 |
| $c_{29}$ (eV nm$^2$) | -0.00672 | 0.00214 | -0.00733 |
| $c_{30}$ (eV nm$^2$) | -0.0558 | -0.0259 | -0.0411 |
| $c_{31}$ (eV nm$^2$) | 0.0285 | -0.0109 | 0.0178 |
| $c_{32}$ (eV nm$^2$) | -0.0581 | -0.0255 | -0.0433 |
| $c_{33}$ (eV nm$^2$) | -0.000342 | 0.00025 | 0.000387 |
| $c_{34}$ (eV nm$^2$) | 0.00537 | -0.00331 | -0.00541 |
| $c_{35}$ (eV nm$^2$) | 0.0223 | -0.0176 | -0.017 |
| $c_{36}$ (eV nm$^2$) | 0.0241 | 0.0197 | -0.0248 |
| $c_{37}$ (eV nm$^2$) | 0.00671 | 0.000335 | -0.00687 |
| $c_{38}$ (eV nm$^2$) | 0.0214 | 0.00371 | -0.00757 |
| $c_{39}$ (eV nm$^2$) | -0.0229 | 0.0034 | 0.0112 |
| $c_{40}$ (eV nm$^2$) | 0.00903 | -0.000739 | 0.00432 |
| $c_{41}$ (eV nm$^2$) | -0.00964 | -0.000377 | -0.0054 |
| $c_{42}$ (eV nm$^2$) | 0.00366 | -0.000194 | 0.00471 |
| $c_{43}$ (eV nm$^2$) | -7.74e-05 | 4.97e-06 | 0.000177 |
| $c_{44}$ (eV nm$^2$) | 0.0266 | 0.0241 | 0.0318 |
| $c_{45}$ (eV nm$^2$) | -0.0156 | -0.00776 | 0.00148 |
| $c_{46}$ (eV nm$^2$) | -0.00304 | -0.00173 | -0.0025 |
| $c_{47}$ (eV nm$^2$) | 0.0326 | 0.0175 | 0.031 |
| $c_{48}$ (eV nm$^2$) | -0.0661 | -0.0482 | -0.0636 |
| $c_{49}$ (eV nm$^2$) | -0.0107 | -0.00713 | -0.0205 |
| $c_{50}$ (eV nm$^2$) | -0.0334 | -0.0158 | -0.0379 |
| $c_{51}$ (eV nm$^2$) | -0.0294 | -0.0139 | -0.0242 |

Table X. Effective Hamiltonian for zincblende crystals considering the $8 \times 8$ extended Kane model.

$$
H_{\mathrm{ZB}} = \left[\begin{array}{cccccccc}
c_0+c_7k^2 & 0 & i\!\left(c_{15}k_-k_z-\tfrac{\sqrt{3}c_5k_+}{2}\right) & -\tfrac{\sqrt{3}i\check{\kappa}c_{12}}{2}-\tfrac{2\sqrt{3}c_{15}k_xk_y}{3}+ic_5k_z & i\!\left(\tfrac{\sqrt{3}c_{15}k_+k_z}{3}+\tfrac{c_5k_-}{2}\right) & i\!\left(c_{15}k_+k_z+\tfrac{\sqrt{3}c_5k_-}{2}\right) & -c_{16}k_xk_y+ic_6k_z & i\!\left(c_{16}k_+k_z+c_6k_-\right)\\[6pt]
0 & c_0+c_7k^2 & \tfrac{ic_{12}\left(2k_z^2-k_\parallel^2\right)}{2} & i\!\left(\tfrac{\sqrt{3}c_{15}k_-k_z}{3}-\tfrac{c_5k_+}{2}\right) & \tfrac{\sqrt{3}i\check{\kappa}c_8}{4}-\tfrac{\sqrt{3}\check{\kappa}c_9}{4}-ic_{13}k_xk_y-ic_5k_z & c_1+\tfrac{c_8\cdot(4k_z^2+k_\parallel^2)}{4}+\tfrac{3c_9k_\parallel^2}{4} & i\!\left(-c_{16}k_-k_z+c_6k_+\right) & c_{16}k_xk_y-ic_6k_z\\[6pt]
i\!\left(-c_{15}k_+k_z+\tfrac{\sqrt{3}c_5k_-}{2}\right) & \tfrac{ic_{12}\left(-2k_z^2+k_\parallel^2\right)}{2} & c_1+\tfrac{c_8\cdot(4k_z^2+k_\parallel^2)}{4}+\tfrac{3c_9k_\parallel^2}{4} & \tfrac{c_{13}k_-k_z-\tfrac{c_3k_+}{2}}{} & -\tfrac{\sqrt{3}c_3k_-}{2} & -c_{13}k_-k_z-\tfrac{c_3k_+}{2} & \tfrac{c_{14}k_+k_z+\tfrac{c_4k_-}{2}}{} & \tfrac{c_{14}k_+k_z+\tfrac{c_4k_-}{2}}{}\\[6pt]
\tfrac{ic_{12}\left(-2k_z^2+k_\parallel^2\right)}{2} & i\!\left(-c_{15}k_-k_z-\tfrac{\sqrt{3}c_5k_+}{2}\right) & -\tfrac{\sqrt{3}c_3k_-}{2} & c_1+\tfrac{3c_8k_\parallel^2}{4}+\tfrac{c_9\cdot(4k_z^2+k_\parallel^2)}{4} & c_{13}k_+k_z-\tfrac{c_3k_-}{2} & \tfrac{\sqrt{3}\check{\kappa}c_8}{4}-\tfrac{\sqrt{3}\check{\kappa}c_9}{4}-ic_{13}k_xk_y-c_3k_z & \tfrac{\sqrt{3}\check{\kappa}c_{10}}{2}-2ic_{14}k_xk_y+c_4k_z & \sqrt{3}\!\left(-c_{14}k_-k_z-\tfrac{c_4k_+}{2}\right)\\[6pt]
\tfrac{\sqrt{3}i\check{\kappa}c_{12}}{2}-\tfrac{2\sqrt{3}c_{15}k_xk_y}{3}-ic_5k_z & i\!\left(-\tfrac{\sqrt{3}c_{15}k_+k_z}{3}+\tfrac{c_5k_-}{2}\right) & \tfrac{\sqrt{3}\check{\kappa}c_8}{4}-\tfrac{\sqrt{3}\check{\kappa}c_9}{4}+ic_{13}k_xk_y+c_3k_z & c_{13}k_-k_z-\tfrac{c_3k_+}{2} & c_1+\tfrac{3c_8k_\parallel^2}{4}+\tfrac{c_9\cdot(4k_z^2+k_\parallel^2)}{4} & -c_{13}k_+k_z-\tfrac{c_3k_-}{2} & c_{11}k^2+c_2 & \tfrac{c_{10}\cdot(-2k_z^2+k_\parallel^2)}{2}\\[6pt]
i\!\left(-c_{16}k_-k_z-c_6k_+\right) & i\!\left(c_{16}k_+k_z-c_6k_-\right) & c_{13}k_+k_z-\tfrac{c_3k_-}{2} & \tfrac{\sqrt{3}\check{\kappa}c_8}{4}-\tfrac{\sqrt{3}\check{\kappa}c_9}{4}+ic_{13}k_xk_y-c_3k_z & \tfrac{\sqrt{3}c_3k_+}{2} & c_1+\tfrac{c_8\cdot(4k_z^2+k_\parallel^2)}{4}+\tfrac{3c_9k_\parallel^2}{4} & 0 & \tfrac{\sqrt{3}\check{\kappa}c_{10}}{2}-2ic_{14}k_xk_y+c_4k_z\\[6pt]
-c_{16}k_xk_y-ic_6k_z & i\!\left(c_{16}k_+k_z-c_6k_-\right) & \tfrac{\sqrt{3}\check{\kappa}c_{10}}{2}+2ic_{14}k_xk_y+c_4k_z & \sqrt{3}\!\left(-c_{14}k_+k_z-\tfrac{c_4k_-}{2}\right) & \tfrac{c_{10}\cdot(2k_z^2-k_\parallel^2)}{2} & -\tfrac{\sqrt{3}\check{\kappa}c_{10}}{2}+2ic_{14}k_xk_y+c_4k_z & c_{11}k^2+c_2 & 0\\[6pt]
c_{14}k_+k_z+\tfrac{c_4k_-}{2} & \sqrt{3}\!\left(-c_{14}k_-k_z+\tfrac{c_4k_+}{2}\right) & c_{14}k_-k_z+\tfrac{c_4k_+}{2} & \sqrt{3}\!\left(-c_{14}k_+k_z+\tfrac{c_4k_-}{2}\right) & \tfrac{c_{10}\cdot(2k_z^2-k_\parallel^2)}{2} & \sqrt{3}\!\left(-c_{14}k_-k_z-\tfrac{c_4k_+}{2}\right) & 0 & c_{11}k^2+c_2
\end{array}\right]
$$

Table XI.  Effective Hamiltonian for wurtzite crystals considering the $10 \times 10$ model with two conduction bands.

$$
H_{\mathrm{WZ}} =
\begin{bmatrix}
c_0+c_{22}k_\parallel^2+c_{44}k_z^2 & ic_9k_- & 0 & c_{33}\!\left(ik_x^2-2k_xk_y-ik_y^2\right) & c_1+ic_{19}k_z+c_{23}k_\parallel^2+c_{45}k_z^2 & k_-\!\left(ic_{10}+c_{37}k_z\right) & k_-\!\left(ic_{11}+c_{38}k_z\right) & c_2+ic_{20}k_z+c_{24}k_\parallel^2+c_{46}k_z^2 & k_+\!\left(ic_{12}+c_{39}k_z\right) & c_{25}\!\left(\hat{K}-2ik_xk_y\right) \\[4pt]
-ic_9k_+ & c_0+c_{22}k_\parallel^2+c_{44}k_z^2 & c_{33}\!\left(ik_x^2+2k_xk_y-ik_y^2\right) & 0 & -k_+\!\left(ic_{10}+c_{37}k_z\right) & c_1+ic_{19}k_z+c_{23}k_\parallel^2+c_{45}k_z^2 & -c_2-ic_{20}k_z-c_{24}k_\parallel^2-c_{46}k_z^2 & k_+\!\left(ic_{11}+c_{38}k_z\right) & c_{25}\!\left(-\hat{K}-2ik_xk_y\right) & k_-\!\left(ic_{12}+c_{39}k_z\right) \\[4pt]
0 & c_{33}\!\left(-ik_x^2+2k_xk_y+ik_y^2\right) & c_{26}k_\parallel^2+c_3+c_{47}k_z^2 & ic_{13}k_- & 0 & 0 & c_{35}\!\left(-ik_x^2+2k_xk_y+ik_y^2\right) & k_-\!\left(-ic_{11}+c_{38}k_z\right) & k_+\!\left(ic_{15}+c_{40}k_z\right) & k_+\!\left(ic_{16}+c_{41}k_z\right) \\[4pt]
c_{33}\!\left(-ik_x^2-2k_xk_y+ik_y^2\right) & 0 & -ic_{13}k_+ & c_{26}k_\parallel^2+c_3+c_{47}k_z^2 & c_{34}\!\left(ik_x^2-2k_xk_y-ik_y^2\right) & c_{34}\!\left(-ik_x^2-2k_xk_y+ik_y^2\right) & 0 & c_{35}\!\left(ik_x^2-2k_xk_y-ik_y^2\right) & k_-\!\left(-ic_{15}+c_{40}k_z\right) & c_{29}\!\left(-\hat{K}+2ik_xk_y\right) \\[4pt]
c_1-ic_{19}k_z+c_{23}k_\parallel^2+c_{45}k_z^2 & k_-\!\left(ic_{10}-c_{37}k_z\right) & 0 & c_{34}\!\left(-ik_x^2-2k_xk_y+ik_y^2\right) & c_{27}k_\parallel^2+c_4+c_{48}k_z^2 & ic_{14}k_- & k_-\!\left(ic_{15}+c_{40}k_z\right) & ic_{21}k_z+c_{28}k_\parallel^2+c_{49}k_z^2+c_5 & k_+\!\left(ic_{16}+c_{41}k_z\right) & c_{29}\!\left(\hat{K}-2ik_xk_y\right) \\[4pt]
k_+\!\left(-ic_{10}+c_{37}k_z\right) & c_1-ic_{19}k_z+c_{23}k_\parallel^2+c_{45}k_z^2 & 0 & c_{34}\!\left(ik_x^2-2k_xk_y-ik_y^2\right) & -ic_{14}k_+ & c_{27}k_\parallel^2+c_4+c_{48}k_z^2 & k_+\!\left(-ic_{15}+c_{40}k_z\right) & -ic_{21}k_z+c_{28}k_\parallel^2+c_{49}k_z^2+c_5 & k_-\!\left(-ic_{16}+c_{41}k_z\right) & c_{29}\!\left(-\hat{K}+2ik_xk_y\right) \\[4pt]
k_+\!\left(-ic_{11}+c_{38}k_z\right) & -c_2+ic_{20}k_z-c_{24}k_\parallel^2-c_{46}k_z^2 & c_{35}\!\left(ik_x^2+2k_xk_y-ik_y^2\right) & 0 & k_+\!\left(-ic_{15}+c_{40}k_z\right) & k_-\!\left(-ic_{15}+c_{40}k_z\right) & c_2+ic_{20}k_z+c_{24}k_\parallel^2+c_{46}k_z^2 & -ic_{17}k_- & k_+\!\left(ic_{43}k_z+c_8\right) & c_{36}\!\left(ik_x^2-2k_xk_y-ik_y^2\right) \\[4pt]
c_2-ic_{20}k_z+c_{24}k_\parallel^2+c_{46}k_z^2 & k_-\!\left(-ic_{11}+c_{38}k_z\right) & k_+\!\left(ic_{11}+c_{38}k_z\right) & c_{35}\!\left(ik_x^2-2k_xk_y-ik_y^2\right) & -ic_{21}k_z+c_{28}k_\parallel^2+c_{49}k_z^2+c_5 & ic_{21}k_z-c_{28}k_\parallel^2-c_{49}k_z^2-c_5 & ic_{17}k_+ & c_{30}k_\parallel^2+c_{50}k_z^2+c_6 & k_-\!\left(-ic_{18}+c_{42}k_z\right) & c_{31}\!\left(\hat{K}-2ik_xk_y\right) \\[4pt]
k_-\!\left(-ic_{12}+c_{39}k_z\right) & c_{25}\!\left(-\hat{K}+2ik_xk_y\right) & k_-\!\left(-ic_{15}+c_{40}k_z\right) & c_{29}\!\left(-\hat{K}+2ik_xk_y\right) & k_-\!\left(-ic_{16}+c_{41}k_z\right) & c_{29}\!\left(-\hat{K}+2ik_xk_y\right) & k_-\!\left(-ic_{43}k_z+c_8\right) & -k_+\!\left(ic_{18}+c_{42}k_z\right) & c_{32}k_\parallel^2+c_{51}k_z^2+c_7 & 0 \\[4pt]
c_{25}\!\left(\hat{K}+2ik_xk_y\right) & k_+\!\left(-ic_{12}+c_{39}k_z\right) & k_-\!\left(-ic_{16}+c_{41}k_z\right) & c_{31}\!\left(\hat{K}+2ik_xk_y\right) & c_{29}\!\left(\hat{K}+2ik_xk_y\right) & c_{31}\!\left(\hat{K}+2ik_xk_y\right) & c_{36}\!\left(-ik_x^2+2k_xk_y+ik_y^2\right) & c_{31}\!\left(\hat{K}+2ik_xk_y\right) & 0 & c_{32}k_\parallel^2+c_{51}k_z^2+c_7
\end{bmatrix}
$$

Table XII. Effective Hamiltonian for rock-salt crystals considering the $4 \times 4$ model composed by the $L_6^{\pm}$ irreps of $D_{3D}$.

$$
H_{\mathrm{RS}} = \begin{bmatrix}
c_0 + c_4 k^2 + c_6 \left( k_x k_y + k_x k_z + k_y k_z \right) & 0 \\
0 & c_0 + c_4 k^2 + c_6 \left( k_x k_y + k_x k_z + k_y k_z \right) \\
-c_2 \left( k_x - k_y \right) - ic_3 \left( k_x + k_y + k_z \right) & c_2 \left( -ik_- + k_z \left( 1 + i \right) \right) \\
c_2 \left( ik_+ + k_z \left( 1 - i \right) \right) & c_2 \left( k_x - k_y \right) - ic_3 \left( k_x + k_y + k_z \right)
\end{bmatrix}
$$

$$
\begin{matrix}
-c_2 \left( k_x - k_y \right) + ic_3 \left( k_x + k_y + k_z \right) & c_2 \left( -ik_- + k_z \left( 1 + i \right) \right) \\
c_2 \left( ik_+ + k_z \left( 1 - i \right) \right) & c_2 \left( k_x - k_y \right) + ic_3 \left( k_x + k_y + k_z \right) \\
c_1 + c_5 k^2 + c_7 \left( k_x k_y + k_x k_z + k_y k_z \right) & 0 \\
0 & c_1 + c_5 k^2 + c_7 \left( k_x k_y + k_x k_z + k_y k_z \right)
\end{matrix}
$$