# Peer review of "DFT2kp: effective kp models from ab-initio data"

_SciPost Physics Codebases_

## Round 1 · Referee Report · Anonymous (Referee 1) · 2023-7-22

Strengths

  1. The paper is well written. It encapsulates the effort of a fully ab initio procedure for modeling k.p Hamiltonians using DFT data from Quantum Espresso package using open source DFT2kp code that is available on github.
  2. The inclusion of practical examples for different materials and structures is well received. These include: graphene, zincblende crystals, wrutzite crystals, rock-salt crystals, and other examples like Bi$_2$Se$_3$ and monolayers of GaBiCl$_2$ and MoS$_2$, respectively. These examples show k.p results for 'all bands Hamiltonian' as well as Hamiltonian for a smaller set of bands treated with Lowdin partitioning. Both cases are compared with DFT results used as input k.p parameters. Significant effort is placed to explain, step-by-step, how to obtain desired group representations for each of the mentioned class of materials. Analytical form of the k.p Hamiltonian in second order of momentum, and numerical values of their parameters are shown in Appendix C.
  3. The code shows much potential as it can easily be applied to both full zone k.p Hamiltonians, using DFT obtained parameters as a good initial guess for fitting, and/or obtaining analytical form and numerical parameters of many k.p Hamiltonians regardless of the crystal symmetry and preferred equivalent basis. Further developments to enable easy input from other DFT codes would be very welcomed.

Weaknesses

  1. The authors acknowledge previous similar work previously done in ref [64] and ref [71], however, they claim that [quote, pg 16, VI. conclusions, last paragraph]: "...here we propose a different method that is more efficient for transformations involving reducible representations, which is necessary when dealing with nearly degenerate bands of different irreps(e.g. spinful graphene)." [end quote] without actual demonstration or enough elaboration for this claim. In sec. III B, only spinless graphene is presented as an example, while an example for spinful grahpine is mentioned in Step 4, it is only referring to an example in the code repository.
  2. In sec. II D. 1.: Modification of the band.x function in Quantum Espresso is explained, that should calculate ${\bf{P}}_{m,n}$ for all $m,n$ and include SOC corrections. However, the motivation for this modification is not elaborated well enough. Was the modification performed just to obtain SOC corrections, or is that way actually faster than using IrRep package to extract QE data to use it with eq (12) and eq (13)? It would be interesting to see in which cases this modification is necessary (i.e. for ${\bf{P}}_{m,n} = \langle m|\pi|n\rangle \approx \langle m|\frac{1}{2} {\bf{v}} | n \rangle $ and when the approximation ${\bf P}_{m,n} \approx \langle m|{\bf{p}}|n\rangle $ is a good approximation.

Report

In general, the paper provides a good review of k.p procedures and steps for obtaining k.p Hamiltonians, which are part of the open source DFT2kp code. It expands upon previous work by suggesting a modification of band.x function to include SOC corrections to momentum ${\bf{P}}_{m,n}$, and provides an alternative procedure to the one in ref [64] and ref [71] that is applicable to reducible representations as well as irreducible. However, I did not find a clear justification for using any of the two mentioned advantages, i.e. including SOC corrections to momentum ${\bf{P}}_{m,n}$ over omitting them, and obtaining unitary transformations for reducible representations over using the same or alternate procedure for irreducible representations. The case were using reducible over irreducible representations would be advantageous, by author's claim, is spinful graphene which is only mentioned in section III B (page 8, Step 4) and again in section VI (page 16, last paragraph in the section) without any analytical or numerical example to support that claim. It is in my opinion that authors should elaborate more on this matter before publishing.

Requested changes

  1. Lowdin partitioning technique is mentioned several times in the paper without any reference to the original paper, for exapmple: In Section II: In the third paragraph (point 3), the fourth paragraph. In Section II A: In the first paragraph, in the fourth paragraph. etc. The reference should contain the paper: https://doi.org/10.1063/1.1748067

  2. In section II: Letter $\alpha$ is used to refer to a set of bands of interest and then again used to refer to fine structure constant. This wouldn't be a problem if two occurrences were not in the same section, so please chose a different label for the set of bands of interest, to avoid possible confusion.

  3. In section II A.:
  4. $V(\bf{r})$ is not defined in equations (1) and (3), and in Appendix A used with the letter $U$ in eq A4. Please define $V({\bf{r}})$ and make it consistent throughout the text.
  5. $\bf{\sigma}$ is not defined, but instead defined in section III in the first paragraph. I suggest to define it on the first occurrence.
  6. In section II B.: I believe there is a mistake in the first paragraph after eq. (6): indices $l,j,k=\lbrace 0,1,2,..\rbrace $ should instead be $i,j,l=\lbrace 0,1,2...\rbrace$. Please correct if this is a mistake.
  7. In sectoin IID. : $V$ is not defined when used in eq (10) but defined in eq (12) and eq (13) as the normalization volume. While it may seem obvious to experienced reader, please avoid repetition of the symbols to avoid possible confusion.
  8. In sec. II D. 1.: Modification of the band.x function in Quantum Espresso is explained, and stated in the same section that "pseudo-wavefunction is a reasonable approximation for the all electron wavefunction, thus neglecting PAW corrections". Perhaps the authors meant "SOC", instead of "PAW" in this case? Please correct me if I am wrong since I am not an expert in PAW calculations.
  9. Section III A.: Figure 1 has too small fonts/sizes to be easily readable for lattice sites in (a) and (b) labeled 1,$\tau$ and $\tau^*$ and legend in (d). I suggest larger fonts or, in overall, a larger Figure. Similar suggestions for Figures 2,3,4, and 5.

  10. Section V A.: Figure 6 has no legend and data is hardly readable in the case of (c) and (d). Please make a visible legend and a larger plot with better formatting. Since coefficients shown in Table I, Table VI, Table VII and Table VIII were not in arbitrary units, the ones in Figure 6 should be in the corresponding units as well, or at least their units should be indicated in the caption or in the legend. I understand that this would make the figure rather large, but since everything is explained well in the text, the figure can be placed in the Appendix.

  11. Section V B:. Third paragraph has, what I think, a grammar error: "... we see that the model does approaches a full zone agreement..." It should say "the model does approach" or "approaches" instead of "does approaches". 5th paragraph: "...which can be used as 'sanity check' for the fitting results..." Perhaps the authors could chose another phrasing of this?

  12. Include spinful graphine as an example (or some other example) to perhaps better illustrate necessity for the usage of the developed method, that is described in section II C, for the symmetry optimization that applies more easily to reducible representations than the ones proposed in ref [64] and ref [71] which uses only irreducible representations.

  13. Further comment, or give a numerical example, on weather the inclusion of SOC correction in ${\bf{P}}_{m,n}$ from section II D.1. can give significantly different results than using approximation from eq (12) and eq (13).

Optional: 11. It is always difficult to show large k.p Hamiltonians on paper. Although the choice of presenting Hamiltonians in tables IX, X and XI as one large matrix is common and intuitive, it can be difficult to read at first. Perhaps if the authors can find a way to separate this into smaller blocks of multiplication tables of irreps contained in the Hamiltonian or, for example, adopt a way to separate in terms of power of momentum $k_i$, $i=\lbrace x,y,z,\parallel,+,- \rbrace $ ?

  • validity: ok
  • significance: good
  • originality: ok
  • clarity: good
  • formatting: acceptable
  • grammar: good

Author:  Gerson J. Ferreira  on 2023-12-15  [id 4197]

(in reply to Report 1 on 2023-07-22)

We thank the first referee for the detailed and overall positive report on our manuscript. The referee report emphasize weaknesses in our manuscript regarding the clarity of two relevant points: the advantages of our methodology to find the unitary transformation between two basis sets; and the necessity and advantages of using the modified bands.x to account for the SOC corrections to the matrix elements. Additionally, the report requests minor changes regarding notation, citations, and clarifications of the two points mentioned above. Below we reply to each of these points.

Weaknesses 1

The authors acknowledge previous similar work previously done in ref [64] and ref [71], however, they claim that [quote, pg 16, VI. conclusions, last paragraph]: "...here we propose a different method that is more efficient for transformations involving reducible representations, which is necessary when dealing with nearly degenerate bands of different irreps(e.g. spinful graphene)." [end quote] without actual demonstration or enough elaboration for this claim. In sec. III B, only spinless graphene is presented as an example, while an example for spinful graphene is mentioned in Step 4, it is only referring to an example in the code repository.

Our reply

We thank the referee for pointing out this weakness in our manuscript. We extended the discussion about this on Section II.C, and added the example of spinful graphene to illustrate this case in detail in Section III.c. The main difference between the methods to find the transformation U is that in the paper by Mozrzymas et al one needs to calculate a weight matrix r_ab identify for which indices (a,b) this matrix is finite. Then, one uses these quantities to calculate U. For transformation between irreps this is straightforward. However, for reducible representations this approach presents subtleties. For concretness, consider that the reducible representation is composed by two irreps G1+G2. In this case we there will be two relevant finite r_ab entries that compose U = U1 + U2. However, there will be a set of r_ab that return a set of equivalent {U1}, and another set of r_ab that return a set of equivalent {U2}. Therefore, in this case one needs to compare the (almost) all matrices in the full set {U1, U2} to identify a pair of nonequivalent U1 and U2. If the reducible representation is composed by a larger number of irreps, the the complexity of the problem increases accordingly. This is a complicated numerical task that can be avoided with our approach.

Weaknesses 2

In sec. II D. 1.: Modification of the band.x function in Quantum Espresso is explained, that should calculate $P_{m,n}$ for all $m,n$ and include SOC corrections. However, the motivation for this modification is not elaborated well enough. Was the modification performed just to obtain SOC corrections, or is that way actually faster than using IrRep package to extract QE data to use it with eq (12) and eq (13)? It would be interesting to see in which cases this modification is necessary (i.e. for $P_{m,n} = \left\langle m | \pi | n \right\rangle \approx \left\langle m | \frac{1}{2} v | n \right\rangle$ and when the approximation $P_{m,n} \approx \left\langle m | p | n \right\rangle$ is a good approximation.

Our reply

We thank the referee for rasing this relvant question. The SOC corrections to Pmn are often neglected in the kp literature, but it can be relevant for materials with strongly localized orbitals, for instance GaN. We have added a discussion about this to Sec. IV.B.1, where we compare the GaN and GaP results with and without the SOC corrections to Pmn.

Requested changes 1

Lowdin partitioning technique is mentioned several times in the paper without any reference to the original paper, for example: In Section II: In the third paragraph (point 3), the fourth paragraph. In Section II A: In the first paragraph, in the fourth paragraph. etc. The reference should contain the paper: https://doi.org/10.1063/1.1748067

Our reply

The referee is absolutely correct and we thank them for pointing out that we missed Löwdin's original manuscript. The reference has been included in the revised version of the paper.

Requested changes 2

In section II: Letter $\alpha$ is used to refer to a set of bands of interest and then again used to refer to fine structure constant. This wouldn't be a problem if two occurrences were not in the same section, so please chose a different label for the set of bands of interest, to avoid possible confusion.

Our reply

Indeed, our notation can cause confusion to the reader. We appreciate the referee for spotting this inconsistency. We have changed sets $\alpha$ and $\beta$ into sets A and B. Additionally, in Sec. II.C the crude DFT basis is now labeled $\mathcal{C}$ and the optimal symmetry-adapted basis is $\mathcal{O}$.

Requested changes 3

In section II A.:
$V(r)$ is not defined in equations (1) and (3), and in Appendix A used with the letter U in eq A4. Please define $V(r)$ and make it consistent throughout the text.

$\sigma$ is not defined, but instead defined in section III in the first paragraph. I suggest to define it on the first occurrence.

Our reply

We thank the referee for spotting these other inconsistencies. We have included the definition of $V(r)$ and $\sigma$ when they first appear in the text.

Requested changes 4

In section II B.: I believe there is a mistake in the first paragraph after eq. (6): indices $l,j,k = \{0,1,2...\}$ should instead be $i,j,l = \{0,1,2... \}$. Please correct if this is a mistake.

Our reply

That is absolutely correct. Thank your for noticing this typo in our text.

Requested changes 5

In section IID.: $V$ is not defined when used in eq (10) but defined in eq (12) and eq (13) as the normalization volume. While it may seem obvious to experienced reader, please avoid repetition of the symbols to avoid possible confusion.

Our reply

We thank the referee for spotting this repetition of $V$. We have correct this inconsistency in the revised version.

Requested changes 6

In sec. II D. 1.: Modification of the band.x function in Quantum Espresso is explained, and stated in the same section that "pseudo-wavefunction is a reasonable approximation for the all electron wavefunction, thus neglecting PAW corrections". Perhaps the authors meant "SOC", instead of "PAW" in this case? Please correct me if I am wrong since I am not an expert in PAW calculations.

Our reply

The sentence is correct, the approximations in eqs. 12 and 13 neglect PAW corrections. But it so happens that the PAW corrections are necessary to consider the SOC. We have modified the sentence in Sec. II.D.1 to emphasize this.

Requested changes 7

Section III A.: Figure 1 has too small fonts/sizes to be easily readable for lattice sites in (a) and (b) labeled 1, $\tau$ and $\tau^*$ and legend in (d). I suggest larger fonts or, in overall, a larger Figure. Similar suggestions for Figures 2,3,4, and 5.

Our reply

We appreciate the referee's keen observation and have increased the font size in the mentioned figures.

Requested changes 8

Section V A.: Figure 6 has no legend and data is hardly readable in the case of (c) and (d). Please make a visible legend and a larger plot with better formatting. Since coefficients shown in Table I, Table VI, Table VII and Table VIII were not in arbitrary units, the ones in Figure 6 should be in the corresponding units as well, or at least their units should be indicated in the caption or in the legend. I understand that this would make the figure rather large, but since everything is explained well in the text, the figure can be placed in the Appendix.

Our reply

We thank the referee for the suggestion. We have improved Fig.6 to match the units given in Tables I, VI, and VII as well as included a proper label for the coefficients in panels (a) and (b). For panels (c) and (d) there are too many coefficients (30) to identify on a figure legend. What matters there is overall trend towards convergence. We have added a sentence to the caption to clarify this. Moreover, in this new figure we have added bottom panels with a single metric to measure the convergence, which is defined by the sum of the discrete derivative the coefficients with respect to the number of remote bands (new eq. 35). This metric provides a better visualization of the convergence. Additionally, revising this figure we have notice that for the 2D materials we were plotting too many coefficients, and now we neglect the coefficients of $k_z^2$, $k_x k_z$, and $k_y k_z$, since for a 2D material there's no dispersion along $k_z$.

Requested changes 9

Section V B:. 
Third paragraph has, what I think, a grammar error: "... we see that the model does approaches a full zone agreement...". It should say "the model does approach" or "approaches" instead of "does approaches".

5th paragraph: "...which can be used as 'sanity check' for the fitting results...". Perhaps the authors could chose another phrasing of this?

Our reply

We thank the referee for spotting these grammar mistakes in our manuscript. We have corrected the text in the revised version.

Requested changes 10

Include spinful graphine as an example (or some other example) to perhaps better illustrate necessity for the usage of the developed method, that is described in section II C, for the symmetry optimization that applies more easily to reducible representations than the ones proposed in ref [64] and ref [71] which uses only irreducible representations.

Our reply

As already mentioned above in the reply to "Weaknesses 1", we have added a discussion about this to Section II.C, and added the spinful example to Sec. III.C.

Requested changes 11

Further comment, or give a numerical example, on weather the inclusion of SOC correction in Pmn from section II D.1. can give significantly different results than using approximation from eq (12) and eq (13).

Our reply

As already mentioned above in the reply to "Weaknesses 2", we have added the GaN and GaP examples to Sec. IV.B.1.

Requested changes 11 (optional)

It is always difficult to show large k.p Hamiltonians on paper. Although the choice of presenting Hamiltonians in tables IX, X and XI as one large matrix is common and intuitive, it can be difficult to read at first. Perhaps if the authors can find a way to separate this into smaller blocks of multiplication tables of irreps contained in the Hamiltonian or, for example, adopt a way to separate in terms of power of momentum $k_i , i = \{ x , y , z , \parallel , + , - \}$ ?

Our reply

We agree with the referee that large matrices are always cumbersome. However, we prefer to keep the current form of large matrices because it matches the forms presented by the QSYMM output in the folder "Examples/PreDefinedModels" of our repository.

---

## Round 1 · Referee Report · Anonymous (Referee 2) · 2023-7-25

Strengths

  1. the code that implements an important methodology was developed
  2. methodology described in detail
  3. examples of application of the code given
  4. limitations of the applicability of the method clearly discussed

Weaknesses

  1. a few places with unclear or potentially misleading statements
  2. use of different terminology than in the existing literature on the problem

Report

In this manuscript, the authors report the development of the code for obtaining k.p Hamiltonians from density functional theory (DFT) calculations, which is available at github. This is an important topic since DFT is too computationally demanding to model realistic nanostructures and k.p is one of the practical methods to do this. However, k.p Hamiltonians have quite a few parameters and it is not easy to obtain the form of the Hamiltonian, nor the corresponding numerical values of the Hamiltonian parameters. As the authors say themselves, the proposed methodology is essentially the same as the methodology of Ref. [64]. On the other hand, the novelty of this manuscript is in the development of the open source code which could be used by interested researchers to obtain the k.p Hamiltonian for the material of interest. For this reason, SciPost Physics Codebases is the right journal for publication of this manuscript.

Overall, the manuscript is clearly written. The methodology is explained in sufficient detail, the examples of the application of the code are given and the limitations of applicability of the code are discussed. Nevertheless, there are a few places in the manuscript where the discussion is not completely clear or the statements made by the authors might be misleading. Therefore, I would suggest the publication of the manuscript after the authors consider the following comments.

Requested changes

1. In page 1, the authors say: "In all cases, the coefficients are typically obtained either (i) comparing the theoretical predictions (band structure, transport, or optical properties) to available experimental data [15, 49], or (ii) fitting to the DFT data [39, 41, 50–53]." I have several remarks regarding this sentence: -It is not clear what is meant by point (i). One can assess the validity of k.p coefficients by comparing the prediction of k.p with experimental data but one can not obtain the k.p parameters that way. -The relation of references [15, 49] to the sentence where these are cited is not clear. Reference [15] is a textbook on spin-orbit effects, while [49] is a long review paper that contains various parameters of semiconducting materials. -There are previous works where k.p coefficients were obtained from DFT data (in addition to Ref. [64] mentioned later in the text by the authors, see for example 10.1016/j.cpc.2007.02.111, 10.1016/j.commatsci.2017.03.017, 10.1103/PhysRevB.78.245107), so I would disagree with the statement that "in all cases" these are obtained by (i) or (ii).

2. In page 1, the authors also say: "For instance, the wannier90 code [54, 55] obtains optimal TB models by fitting the DFT data." -I think that this statement is not correct. The main purpose of the wannier90 code is to construct the localized Wannier functions. To my knowledge, there is an option in the code to use these wave functions to calculate the Hamiltonian matrix elements in the basis of these wave functions (which can further be used to obtain the band structure at arbitrary point in the Brillouin zone). This procedure does not involve any kind of fitting.

3. In page 4, the authors say: "Here, however, we propose an alternative method that applies more easily to reducible representations." It remains unclear what is the relevance of reducible representations. The eigenstates of the Hamiltonian transform in accordance with irreducible representations of the symmetry group at a given point in the Brillouin zone.

  1. At the end of Section II C the authors report a residuum minimization procedure to obtain the parameters of the matrix U. It remains unclear what guarantees the uniqueness of the result obtained (up to an overall phase).

  2. In Section V B the authors discuss if the k.p method with many bands could lead to an accurate description of the band structure throughout the whole Brillouin zone. Although this should be possible in principle by taking infinitely many bands, they find in practice that a full zone description is not reached with practically achievable number of bands. I fully believe that this conclusion is correct. However, I do not agree with the authors' suggestion to use the parameters obtained from their procedure as an initial guess for additional fitting of the band structure. The fact that full zone description cannot be reached using the procedure described in this paper implies that fitting parameters obtained in other works do not have a physical meaning and that these cannot therefore be used to make any other predictions except as a fit to the band structure.

  3. Given that the code developed essentially implements the methodology of Ref. [64], there are many differences in the terminology, i.e. the authors of this manuscript use quite a different terminology than in Ref. [64] which was published three years ago. Here are a few examples:

this manuscript: Ref. [64]:

optimal basis symmetry-adapted basis Hamiltonian shape Hamiltonian form crude basis DFT basis

For the benefit of the reader, the authors should connect their terminology with existing terminology in the literature or adopt the terminology previously used in the literature.

  • validity: high
  • significance: good
  • originality: good
  • clarity: good
  • formatting: excellent
  • grammar: good

Author:  Gerson J. Ferreira  on 2023-12-15  [id 4196]

(in reply to Report 2 on 2023-07-25)
Category:
answer to question

We thank the second referee for the detailed and overall positive report on our manuscript. The referee report points towards a couple of misleading statements that indeed needed to be corrected, and questions about the motivation for our method to find the transformation U. Additionally, the report requests minor changes regarding notation. Below we reply to each of these points.

Weaknesses 1

  1. a few places with unclear or potentially misleading statements

  2. use of different terminology than in the existing literature on the problem

Our reply

We appreciate the referee for pointing out the weaknesses of our manuscript. In the revised version, we have improved the text to eliminate these weak points.

Requested changes 1

In page 1, the authors say: "In all cases, the coefficients are typically obtained either (i) comparing the theoretical predictions (band structure, transport, or optical properties) to available experimental data [15, 49], or (ii) fitting to the DFT data [39, 41, 50–53]." I have several remarks regarding this sentence: -It is not clear what is meant by point (i). One can assess the validity of k.p coefficients by comparing the prediction of k.p with experimental data but one can not obtain the k.p parameters that way. -The relation of references [15, 49] to the sentence where these are cited is not clear. Reference [15] is a textbook on spin-orbit effects, while [49] is a long review paper that contains various parameters of semiconducting materials. -There are previous works where k.p coefficients were obtained from DFT data (in addition to Ref. [64] mentioned later in the text by the authors, see for example 10.1016/j.cpc.2007.02.111, 10.1016/j.commatsci.2017.03.017, 10.1103/PhysRevB.78.245107), so I would disagree with the statement that "in all cases" these are obtained by (i) or (ii).

Our reply

We apreciate the referee's comment for pointing out the lack of clarity in our text as well as the additional references. These new references actually implement a k.p directly from DFT calculations, as the referee mentions. We have rephrased this sentence in order to convey our idea more clearly and incorporate these new references. We have substantially modified the 4th paragraph of our introduction to accomodate the referee's comment.

Requested changes 2

In page 1, the authors also say: "For instance, the wannier90 code [54, 55] obtains optimal TB models by fitting the DFT data." -I think that this statement is not correct. The main purpose of the wannier90 code is to construct the localized Wannier functions. To my knowledge, there is an option in the code to use these wave functions to calculate the Hamiltonian matrix elements in the basis of these wave functions (which can further be used to obtain the band structure at arbitrary point in the Brillouin zone). This procedure does not involve any kind of fitting.

Our reply

The referee is absolutely correct. The wannierization approach does not implement any fitting but rather obtain the hopping parameters by direct calculations. We have corrected this statement in the revised version.

Requested changes 3

In page 4, the authors say: "Here, however, we propose an alternative method that applies more easily to reducible representations." It remains unclear what is the relevance of reducible representations. The eigenstates of the Hamiltonian transform in accordance with irreducible representations of the symmetry group at a given point in the Brillouin zone.

Our reply

This question was also raised by the first referee. To clarify this we have extended the discussion about this on Section II.C, and added the example of spinful graphene to illustrate this case in detail in Section III.c. The referee here is correct that ideally the eigenstates should transform in accordance with the irreps, however, one must keep in mind that the DFT/QE eigenstates are obtained numerically via iterative methods (e.g., Davidson algorithm), and for nearly degenerate irreps there can always be residual, yet relevant, mixing between the eigenstates. Indeed this is the case for spinful graphene, for which the gap between the irreps is due to the intrinsic SOC. For graphene this intrinsic SOC is quite small.

Requested changes 4

At the end of Section II C the authors report a residuum minimization procedure to obtain the parameters of the matrix U. It remains unclear what guarantees the uniqueness of the result obtained (up to an overall phase).

Our reply

The referee is absolutely correct. The minimization procedure raises questions about the uniquiness of the solution. We have significantly changed the discussion at the end of Section II.C to properly discuss this issue. At the global minima of the residues R, the solution is not unique, since it allows for arbitrary relative phases between the irreps that compose a possibly reducible representation of the basis functions (neglecting a possible global phase). There could also be local minima at finte R, but notice that the global minima occurs necessarily for R=0. So we can use the value found for residue R to verify if we are at the global miminia (R close enough to zero) or at a local minima (R=0). In the next version of the code we shall print this value as a report to the user with possible warinings if R is not zero.

Requested changes 5

In Section V B the authors discuss if the k.p method with many bands could lead to an accurate description of the band structure throughout the whole Brillouin zone. Although this should be possible in principle by taking infinitely many bands, they find in practice that a full zone description is not reached with practically achievable number of bands. I fully believe that this conclusion is correct. However, I do not agree with the authors' suggestion to use the parameters obtained from their procedure as an initial guess for additional fitting of the band structure. The fact that full zone description cannot be reached using the procedure described in this paper implies that fitting parameters obtained in other works do not have a physical meaning and that these cannot therefore be used to make any other predictions except as a fit to the band structure.

Our reply

We certainly agree with the referee regarding the physical meaning of the fitted parameters. For some cases, the fitted results may lead to unrealistic physical parameters. We have adapted the text convey this information effectively, and we suggest that our calculated values can provide an important benchmark to verify are close to realistic parameters.

Requested changes 6

Given that the code developed essentially implements the methodology of Ref. [64], there are many differences in the terminology, i.e. the authors of this manuscript use quite a different terminology than in Ref. [64] which was published three years ago. Here are a few examples:

this manuscript: Ref. [64] optimal basis: symmetry-adapted basis Hamiltonian shape: Hamiltonian form crude basis: DFT basis

For the benefit of the reader, the authors should connect their terminology with existing terminology in the literature or adopt the terminology previously used in the literature.

Our reply

We agree with the referee that we must adapt our terminology. In the revised version we use the notation: optimal symmetry-adapted basis; Hamiltonian form. Additionally, for the term "crude basis" we now use "crude DFT basis" to emphasize that it refers to the untouched DFT basis, while the "optimal symmetry-adapted basis" is still obtained from DFT, but rotated to properly match the symmetry representations. The revised text emphasizes that the crude basis refer to the untouched basis extracted directly from DFT.

---

## Editorial Decision

resubmitted